# Building a methodological framework and toolkit for news media dataset tracking of conflict and cooperation dynamics on transboundary rivers

Liying Guo[1], Jing Wei[1], Keer Zhang[1], Jiale Wang[1], Fuqiang Tian[1]*

[1]Department of Hydraulic Engineering, State Key Laboratory of Hydroscience and Engineering, Tsinghua University, Beijing, 100084, China

*Correspondence to*: Fuqiang Tian (tianfq@tsinghua.edu.cn)

**Abstract.** Management of transboundary rivers will be one of the great political and environmental challenges of the 21st century if knowledge of conflict and cooperation is not fully developed. Transboundary river conflict and cooperation are critical for the sustainable development of river basins, regional security, and stability, and have significant scientific and practical implications. The construction of a dataset of transboundary water events – individual conflictive or cooperative interaction between riparian –provides important data support and factual basis for the study of transboundary rivers. However, the most representative research, the Transboundary Freshwater Dispute Database, is built by means of manual reading for information extraction, thus difficult for fast updating, also does not cover the global changes in the past decade. This research aims to build a methodological framework for news media datasets tracking of conflict and cooperation dynamics on transboundary rivers, provide mass of relevant data for the research of transboundary rivers in the globe, prepare a potent research toolkit, lay a solid foundation for further data mining research, and better suit the big data age. In order to test the effectiveness of the methodological framework and toolkit for dataset construction, this research analyses the spatial coverage both in terms of continental and national, temporal coverage from 1953 to 2019, content coverage and conducts relevance screening of the articles in the four representative river basins in the datasets. The results show that the datasets built by this framework can capture comprehensive contents of transboundary water conflict and cooperation in both spatial and temporal coverage with acceptable data quality.

## 1 Introduction

Globally, there are 310 transboundary river basins, covering 47.1% of the land area except Antarctica (McCracken & Wolf, 2019), and accounting for approximately 60% of global freshwater discharge (Wolf et al., 1999). The population of the basins comprises 52% of the world's total (McCracken & Wolf, 2019). Transboundary river basins not only support the lives of the people in the basins, but also connect the various economic sectors and ecosystems in the basin into an organic whole; transboundary water management not only affects the development of riparian countries in all aspects, but also intertwines social, economic, environmental, and political sectors of each riparian country and increase interdependence in between

(United Nations, 2019). Riparian countries have divergent demands and priorities for transboundary water resources, different development agendas for water resources, and different water governance regimes and water resources cultures (Sadoff & Grey, 2005), which make the management of transboundary water resources more complex than that of domestic water resources. Transboundary river basins are thus prone to conflicts of various forms, forming a complex situation where conflicts and cooperation develop intertwined. Therefore, research on water conflict and cooperation in transboundary rivers has important theoretical value and practical significance. Exploration of dynamics of conflict and cooperation as social sectors in a human-water coupled transboundary system is especially prominent.

Among the extant studies on transboundary rivers, transboundary water event datasets – individual conflictive or cooperative interactions between riparian – provide factual data support for the formation of global generalized understanding, which is of great significance. The most representative research - Transboundary Freshwater Dispute Database (TFDD) developed by Oregon State University (Wolf, 1999) has compiled more than 6,400 historical transboundary water events, both conflictive and cooperative (3813 left after removing duplicated records by us from their original data) on the global scale from the year of 1948 to 2008 (Transboundary Freshwater Dispute Database, 2008). The data came from existing political science datasets and news media articles, which was manually screened, interpreted, and coded to extract the detailed information of the water event (Yoffe & Larson, 2001). Building upon these event data, Basin at Risk Projects (BAR) (Yoffe & Larson, 2001) further classified water events by level of intensity of conflict or cooperation, ranging from -7 to +7 to identify potential socio-political threats, and provided a brief summary of the detailed information of the event. The results included very few examples of full cooperation and extreme conflicts but identified river basins that are at potential risk for further conflict. TFDD has built up foundation of this methodological framework for tracking transboundary river water events and allows for further identification of the conflict/cooperation dynamics and possible analysis of its complex driving mechanism.

Given that manual reading and coding processes was adopted in TFDD, which largely limit the implication of this method in the era of big data. The explosion of digital news data, whose discussion of transboundary water events has grown exponentially, made it more difficult to manually track all published water events and the dynamics of conflict and cooperation. While manual reading excels in extracting latent and detailed content, it is much more time and labor consuming. Therefore, it is necessary to revise the methodological framework to meet the current need for a more comprehensive and detailed dataset which can be updated in a more efficient manner. Meanwhile it can also provide the basis for further analysis, i.e. to reflect the concerns of different stakeholders, obtain a global law of transboundary water conflict and cooperation (Bernauer & Böhmelt, 2020).

This paper aims to provide such a revised methodological framework for news media tracking of conflict and cooperation dynamics on transboundary rivers and provide a toolkit when applying the framework in the corresponding research. The theory that inspired our framework is from Lasswell's model of communication (Lasswell, 1948), who focused on communication as a process, to conduct problem-oriented inquiry of the news report through content analysis with the seven fundamental elements "who, with what intentions, in what situations, with what assets, using what strategies, reaches what audiences, with what result?". Our design of search keywords generator follows closely to the line of theoretical principles by

Lasswell and intends to track conflict and cooperation dynamics on transboundary rivers by answering Lasswell' question
involved with seven elements. This study can help to reveal the evolutionary dynamics and patterns of transboundary water
conflicts and cooperation on a global scale, collecting news media datasets with an automated approach, and minimizing the
manual workload of screening, reading and understanding the relevant news media articles, and provides researchers with
powerful tools to retrieve useful information in related fields. It can serve as the foundation for further analysis via text mining
and as a methodological foundation of quantifying the social dimension of transboundary river systems.

## 70   2 Data and Method

This study attempts to build a revised methodological framework that reflects the dynamics of water conflicts and cooperation
among all the transboundary rivers in the globe. Overall procedures in the revised framework are illustrated in Figure 1. The
method can be divided into three steps: Step 1 Select Database, Step 2 Keyword Determinants, Step 3 Data Cleaning and
Processing. More specifically, the method begins with selecting news database in Step 1, detailed criteria to select news
databases is stated in Sect. 2.1.1. Search keywords are generated in Step 2 with 5 blocks of keywords determinants. These 5
blocks concern with river basin characteristics and the research question and determine the validity and relevance of the data
to be collected. Using generated keyword in Step 2, original dataset is downloaded for data cleaning and processing in Step 3,
which include rough manual reading and sorting to check results relevance in order to feedback on further keywords
modification in Step 2. Trial-and-errors between Step 2 and Step 3 promise satisfactory keywords setting for the research. In
Sect.2.4, several potentials for analysis in the future are introduced, which are extended applications for this methodological
framework.

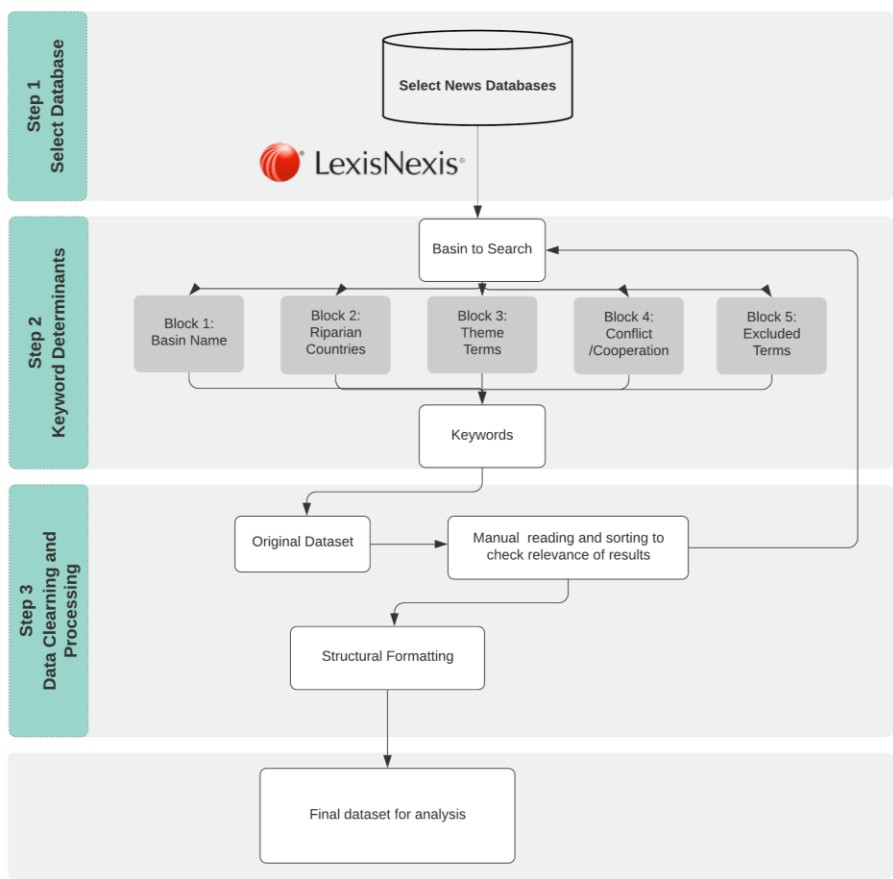


**Figure 1.** Method flow chart

**2.1 Step 1: Select Database2.1.1 News Media as Data Source**
Choice of media sources should accord closely with the research goal. Our research goal is to track conflict and cooperation
dynamics on transboundary rivers, which requires the data to cover water events and public opinion in a relatively long period
of time. Also, newspapers (both print news and web news) published by professional journalists and editors are more suitable
to use as data sources to reflect opinions of communities than social media (e.g., Twitter) as reflections of individual opinions.
News media reflect what is important for the individual country/sector they are published within (Cooper, 2005), it thus has
increasingly been studied by researchers to gain insight into transboundary water issues. The local news media is the first-hand
material that reflect attitude/perception riparian countries held for their shared water and the involved stakeholders when
discussing the water events in the transboundary river basin. In parallel, international news media serve as a good source of
information to understand viewpoints from international audience that are outside of the river basins. Together, text analysis
of both regional and international news for water events in transboundary rivers can reveal the full picture of the ongoing
dynamics in the river basin.
**2.1.2 Select News Database**
The very first step of this method involve selecting a news database that covers comprehensive news sources spanning across
the globe. The selected media databases should include longitudinal coverage (i.e., can be traced back to decades) and updated
in a timely manner, such as Lexis Advance (a product of Lexis Nexis Corporation), ProQuest, Factiva, etc. Lexis Advance
covers more than 6,000 mainstream news media in most countries and regions around the world with a long-term coverage,
and is one of the most commonly used news sources in the field of social sciences (Weaver & Bimber, 2008; Racine et al.,
2010). Therefore, Lexis Advance is taken as an example of news media database to demonstrate the process of obtaining news
media data of transboundary water conflicts and cooperation, and other suitable databases can, of course, be feasible options.
Although the temporal coverage is affected by the level of media development in different regions, the covered timeframe
spans over one hundred years to date, providing good data support on tracking media coverage of transboundary water conflict
and cooperation research. The scope of research limit to English newspaper only due to our limitation of language processing,
which is considered as sufficient enough to meets the requirements for extensive coverage of transboundary water conflicts
and cooperative research.
**2.2 Step 2: Keyword Determinants**
**2.2.1 Select Rivers to Search**
The scope of rivers to search in this study are 286 transboundary rivers as identified in 2016 (Transboundary Waters
Assessment Programme, 2016). It is understood that the total number of transboundary rivers are recently been updated to 310
(McCracken & Wolf, 2019), which are due to advancement of remote-sensing technology. Remote sensing can examine the
two fundamental characteristics of transboundary rivers (common terminus and perennial), thus finer resolution of hydrologic
data assists in discovering new transboundary rivers. In general, majority of the 24 newly added basins are small in area (less
than 10,000 km2) (McCracken & Wolf, 2019), and are considered as inactive in conflicts and cooperation dynamics. Therefore,
this study holds on 286 transboundary rivers in the procedure of Select Rivers to Search, which can be extended to 310 in the
future. Four river basins were taken as case studies, Mekong, Nile, Columbia, and Ganges-Brahmaputra-Meghna (hereafter as
GBM) as the global hotspot of water events.
**2.2.2 Search Keywords Generator**
The search terms are one of the key determinants of the coverage and relevance of the data to be retrieved. This study develops
a keyword generator that allow efficient generating of keywords terms, which are applicable to all transboundary river basins
(286 rivers basins) in the world. The keyword determinants are developed on the basis of TFDD (Yoffe & Larson, 2001) and
further revised to include five blocks of terms (as shown in Figure 2). These five blocks aim to include in which river basin
(Block 1), who (riparian countries, Block 2), regarding what issues (Block 3), have resulted in Conflict/Cooperation status
(Block 4). More specifically, Block 1 and Block 2 are basic information about the river basin, such as name of the river basin,
and various formats of riparian countries' names, retrieved articles need to discuss the conflictive or cooperative aspects of the
events involving at least one of riparian countries; Block 3 contains theme terms regarding of various functions of the water
body, topics discussing hydraulic infrastructure, water quality, agriculture/fishing, or any other specific topics with associated
terms; Block 4 include keywords indicate conflict or cooperation; and Block 5 consist of keywords to be excluded which bring
in irrelevance. The above five blocks can narrow down the search to the desired scope, with the list of unwanted words further
screen out irrelevant topics, after which, the search results can achieve a balance between coverage and relevance, that is,
neither too much relevant information is missed, nor too much irrelevant information is included.

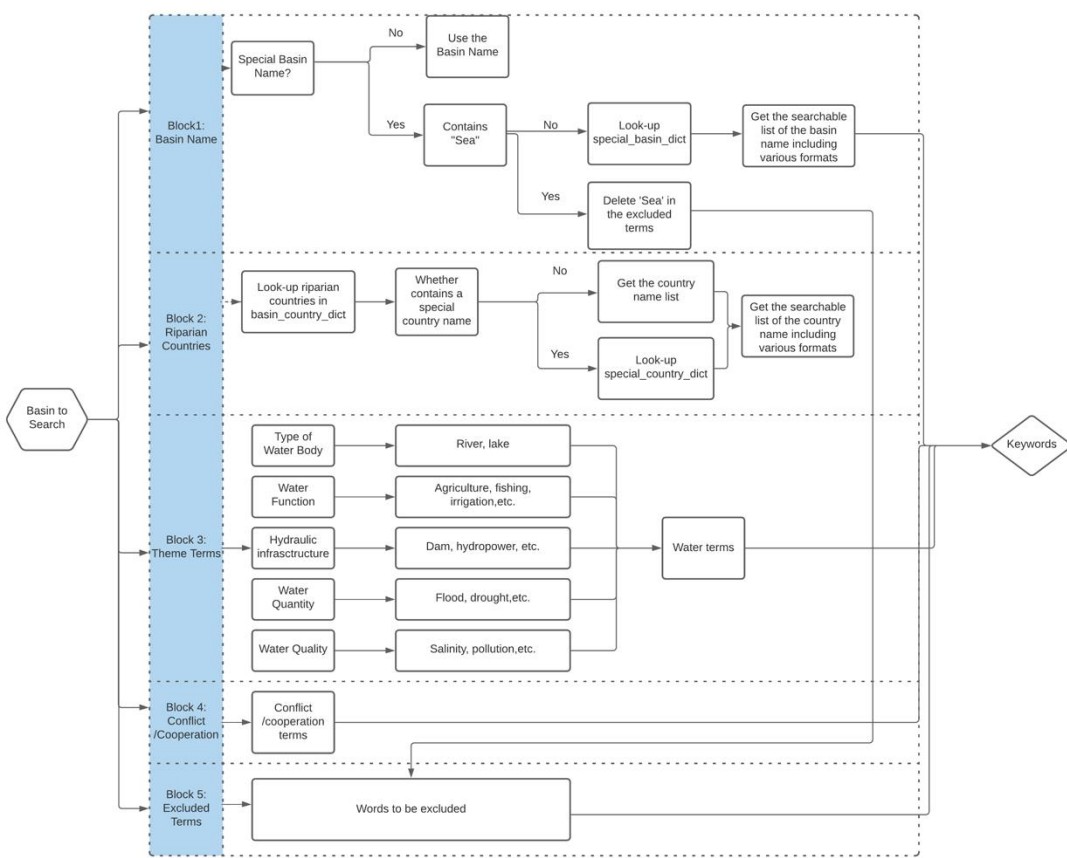

**Figure 2.** Search Keywords Generator flow chart
*(1) Block 1: Basin Name*
This study customizes relatively general algorithms to generate search strings for river basins with different attributes and
conducts special treatments for individual river basins, so that each river basin is under the general search rules resulting in a

considerable number of search results with a balance of coverage and accuracy. The aim of ***Block 1*** is to get the searchable list of the basin name including various formats and consider special treatments for specific categories of basin names. There are several categories identified for different variations of basin names, see below for specific information.

a) Basin name same as the name of a certain riparian country or state; the search results are likely to contain many articles about the internal affairs and diplomacy of the country or state. The detailed list of this type of basins is shown in Table 1. When talking about transboundary water issues, people usually focus on interactions on the scale of local communities and riparian states rather than intercontinental, and do not refer to the Continent names. Therefore, raising the frequency of continent name in search keywords will only compress data volume of relevant articles significantly, but not improve the data relevance pertaining to the research goal. However, river basins with the same names but located in different continents have different riparian countries. Adding frequency setting of riparian countries will filter out articles about the river on the other continent effectively. For example, St. John rivers appear both in Africa (flowing through Côted'Ivoire, Guinea, and Liberia) and North America (flowing through the United States and Canada). Rising frequency of riparian countries rather than continent names contributes more to the data relevance.

b) Basin name contains commonly used words, for example, Amazon, which not only refers to the Amazon river basin, but also an e-commerce company in the United States. More filters will be adopted in this case to ensure relevance rate. See Table 1 for a detailed list of this type of river basins.

c) Basin name contains words such as 'Lake' or 'Sea', the word frequency setting for 'River' in the search string needs to be modified, and that for 'Lake' needs to be increased, or 'Sea' needs to be removed from the list of noise keyword. See the detailed list of this type of river basin in Table 1.

d) Other categories of basin names that require special treatment (see Table 1 for details) are: river basins have different names, such as upstream and downstream rivers are designated with different names, or the river basin contains multiple rivers; rivers in the basin have different names; the basin name is composed of multiple words; similar basin names exist on different continents; the basin name contains 'St.', but may be referred as 'Saint' in media articles.

**Table 1.** Categories of basins need special treatment

| Categories of basins need special treatment | Basin names | Treatment |
|---|---|---|
| Basin name includes state's or district's name | Belize; Columbia ; Congo/Zaire ; Corredores/Colorado ; Gambia ; Jordan ; La Plata ; Mississippi ; Nelson-Saskatchewan ; Niger ; Senegal ; Tigris-Euphrates/Shatt al Arab | Raise the frequency setting for 'water' or 'river' etc. to filter out the geopolitical articles as many |
| Basin name includes common word | Amazon; Baker; Columbia; Cross; Don; Fly; Han; Lagoon Mirim; Lotagipi Swamp; Massacre; Negro; Oral/Ural; Orange; Rhone; Red/Song Hong; San Martin; Seno Union/Serrano; Vanimo-Green; Whiting | Raise the frequency setting for 'water' or 'river' etc. to filter out water-unrelated articles as many; or delete a certain percentage of articles from the end of the results list |
| Basin name includes 'Lake', 'Sea' | Lake Chad; Lake Fagnano; Lake Natron; Lake Prespa; Lake Titicaca-Poopo System; Lake Turkana; Lake Ubsa-Nur; Aral Sea | The word frequency setting for 'River' in the search string needs to be modified, and that for |

| | | |
|---|---|---|
| | | 'Lake' needs to be increased, or 'Sea' needs to be removed from the list of noise keywords |
| Basin name includes multiple formats (maybe consists of multiple rivers) | Asi/Orontes; BahuKalat/Rudkhanehye; Bei Jiang/Hsi; Benito/Ntem; Ca/Song-Koi; Cancoso/Lauca; Carmen Silva/Chico; Coco/Segovia; Congo/Zaire; Corantijn/Courantyne; Corredores/Colorado; Cuvelai/Etosha; Douro/Duero; Gallegos/Chico; Ganges-Brahmaputra -Meghna; Hamun-i-Mashkel/Rakshan; Hari/Harirud; Ili/Kunes He; Jenisej/Yenisey; Juba-Shibeli; Kura-Araks; Lava/Pregel; Mana-Morro; Nelson-Saskatchewan; Oder/Odra; Oiapoque/Oyupock; Oral/Ural; Red/Song Hong; Seno Union/Serrano; Shu/Chu; Tagus/Tejo; Tigris-Euphrates/Shatt al Arab; Tjeroaka-Wanggoe; Torne/Tornealven; Vanimo-Green; Vistula/Wista | Contain all formats of related basin/river names in the search keywords |
| Basin name consists a river with multiple names | Muhuri (aka Little Feni) | Contain all formats of related river names in the search keywords |
| Basin name includes multiple words | An Nahr Al Kabir; Astara Chay; Coatan Achute; El Naranjo; Great Scarcies; Har Us Nur; Kowl E Namaksar; La Plata; Lagoon Mirim ; Lotagipi Swamp; Lough Melvin; Nahr El Kebir; Oued Bon Naima; Pu Lun T'o; Rio Grande (N. America); Rio Grande (S. America); San Martin; Song Vam Co Dong; St. Croix; St. John (Africa); St. John (North America); St. Lawrence; St. Paul; Wadi Al Izziyah | Add quotation mark to the basin name in the search keywords to search it as a whole, and prevent the basin name tokenized |
| Same basin names exist in multiple continents | Great/Little Scarcies; Rio Grande (N. America/S. America); St. John (Africa/North America) | Usually, articles do not contain the continent name when talking about rivers. Therefore, adding continent names into search keywords compresses data volume significantly and does not help with relevance. Adding frequency setting of riparian countries will filter out articles about the river on the other continent effectively. |
| Basin name includes St. (Saint) | St. Croix; St. John (Africa); St. John (North America); St. Lawrence; St. Paul | Put 'saint' and 'St.' into search keywords together |

The *special_basin_dict* in the toolkit in *Block 1* is a python dictionary uploaded on Zenodo, whose *keys* are basin names with
multiples words, or with special characters (e.g., back slash, dash, or parenthesis), and *values* are all searchable formats of the
related basin names and river names. Given the original basin name to search, *special_basin_dict* can feedback its
corresponding searchable keywords. If without *special_basin_dict* and using the original basin name to search, few results
even none can be found. Coverage of retrieved results is enhanced by the *special_basin_dict*. When using the dictionary,
import it to your script first, and call it easily.
*(2) Block 2*
*Block 2* is information concerning with riparian countries within the transboundary river basin. The aim of *Block 2* is to get
the searchable list of the riparian country names including various formats. To fulfill the task, two helpful dictionaries -
*basin_country_dict* and *basin_country_dict* are developed and provided in the toolkit of this study.
The *basin_country_dict* in the toolkit in *Block 2* is a python dictionary uploaded on Zenodo, whose *keys* are basin names, and
*values* are all riparian countries located in the transboundary basin. Given the basin name to search, *basin_country_dict* can

feedback the list of riparian countries. Another python dictionary used in *Block 2* is *special_country_dict*, whose *keys* are country names with various formats, or with special characters (e.g., dot), *values* are all the searchable formats of the country name. Given the special country name to search, *special_country_dict* can feedback the list of all searchable formats of the country name.

Given a basin name to search, first looking up riparian countries in the *basin_country_dict* gets the list of riparian countries; then check whether there is a special country name in the list of riparian countries. If yes, through looking up *special_country_dict*, all searchable list of the country name including various formats are generated in *Block 2*.

*(3) Block 3*

*Block 3* contains terms concerning various themes of transboundary water resources, shown in Table 2. For example, type of water body, function of water body (agriculture, fishing etc.), hydraulic infrastructure, water quantity, water quality, and other specific topics which arouse certain research interests.

*(4) Block 4*

*Block 4* contains conflict/cooperation related keywords, adopted from TFDD searching keywords (Yoffe & Larson, 2001), shown in Table 2. If you focus on a certain type of conflict/cooperation, keywords in *Block 4* can be modified accordingly. In addition, UNBIS Thesaurus (UNBIS Thesaurus, 2021) provides lists of related keywords for conflict and cooperation which can be referred to.

*(5) Block 5*

*Block 5* contains excluded terms, given the research goal of our study, most of which are adopted from TFDD searching keywords (Yoffe & Larson, 2001), shown in Table 2. These terms, seemingly relevant to our topics, occur in media articles massively and easily bring in lots of data noise. For example, 'sea' and 'ocean' bring mass of irrelevant articles talking about marine rights and navigational utilization; 'nuclear' refers to 'nuclear power' and 'nuclear threaten', which is not the main concern of transboundary water conflict and cooperation; and as for 'flood of refugees', though it contains the keyword 'flood', but is regarded as irrelevant to our topics. These terms prone to bring in noise should be excluded in searching results, and thus list in excluded terms in *Block 5*. If researchers employ our framework in their own study fields in the future, excluded terms to avoid noise in Block 5 should be modified accordingly to fit their own research field based on results of trial-and-error between Step 2 and Step3 and combined with their experience and knowledge background. For example, when collecting data for Aral Sea, 'sea' should be deleted from the excluded terms in *Block 5* to prevent great loss of data coverage.

**Table 2.** Example of keywords in Block 1-5

| Block 1: Basin name | Basin name (5) | |
| Block 2: Riparian countries | Each riparian country (2) | |
| | **Type of water body** | Water (3), river (3), lake, stream, tributary, etc. |

| Block 3: Theme terms of transboundary water resources | Function of water body | Irrigation, fish, fish rights, water rights, water diplomacy, water hegemony, etc. |
| | Hydraulic infrastructures | Dam, diversion, channel, canal, hydroelect*, hydropower, reservoir, etc. |
| | Water quantity | Flood, drought*, water allocation, water sharing, etc. |
| | Water quality | Salinity, pollution, etc. |
| Block 4: Conflict/cooperation terms | Conflict | dispute*, conflict*, disagree*, war, troops, "letter of protest", hostility, "shots fired", boycott, protest* |
| | Cooperation | Treaty, agree*, convention, "framework directive", negotiat*, resolution, commission, secretariat, "joint management", "basin management", peace, "accord", "peace accord", settle*, cooperat*, collaborat*, bilateral, multilateral, sanction* |
| Block 5: Excluded terms | | Sea, ocean, navigat*, nuclear, water cannon, light water reactor, mineral water, hold water, cold water, hot water, water canister, water tight, water down*, flood of refugees, oil, drugs, a stream of, flood of |

Notes: asterisk (*) indicates root of a word; number in parentheses (5,2 or 3) indicate at least how many times the keywords should appear in a searching result

### 2.2.3 Term frequency setting of keywords

The setting of term frequency of keywords comes from the recursive trial-and-errors in the search process, which makes the search results for most transboundary river basins relatively satisfactory. For individual river basins, universal setting rules of term frequency will cause the search results drop to zero sharply or too many to cope with, and the accuracy of the search results cannot be guaranteed. For example, when collecting data on the Jordan River Basin, given that Jordan is not only the name of the river basin, but also the name of a riparian country in the basin, there are too many articles that meet all the search requirements but purely about regional politics. Therefore, the setting of term frequency for the keywords 'water' and 'river' needs to be increased to 5 times to highlight the theme of transboundary water resources and ensure that the search results have similar accuracy to other river basins.

Taking the Lancang-Mekong basin as an example, the search keywords used in this study are shown in Table 3. During the trial-and-error process, we found that the results relevance rate is far below acceptable level (less than 30%), therefore we revised the keyword terms to increase frequency of certain terms until satisfactory results are produced, for example, the name of the basin appears in the article were increased to at least five times, the name of any riparian country in the basin (official name or abbreviation) appears in the article at least two times. Water-related words are divided into three sub-blocks: type of water body, function of water body, and infrastructures for water conservancy. Among them, 'water' and 'river' appear at least 3 times respectively, and the rest keywords of water block appear at least once; words related to conflict, or cooperation appear at least once. Recordings of trial-and-error process for Mekong, Nile and Jordan River Basin are provided in Appendix to demonstrate the effects of various groups of frequency settings of keywords and how balance between relevance and coverage is approaching. Although term frequency settings of keywords and justification of balance between relevance and coverage in this study may not be optimal, with a certain degree of coexisting subjectivity and objectivity, they can also serve as a reference for other researchers.

**Table 3.** Search Keywords in the study (Lancang-Mekong as an example)

| Key Word Search | Lexis Advance Database |
|---|---|
| **Must Include the Basin Name (at least 5 times)** | Mekong (5) |
| **Includes at least one of the following countries' name (at least twice)** | Thai*(2), Cambodia*(2), China(2), Chinese(2), Laos(2), Myanmar(2), Burm*(2), vietna*(2) |
| **Includes at least one of the following words related to Water** | Same as Block 3 (see Table 2) |
| **Includes at least one of the following words related to Conflict/Cooperation** | Same as Block 4 (see Table 2) |
| **Does not include any of the following noisy words** | Same as Block 5 (see Table 2) |

Notes: asterisk (*) indicates root of a word; number in parentheses (5,2 or 3) indicate at least how many times the keywords should appear in a searching result

## 2.3 Step 3: Data Cleaning and Processing

Before finalizing the refined datasets for further analysis, data cleaning and processing is indispensable. The first stage in Step 3 is Rough Manual Reading and Sorting to Check Results Relevance, which aims to provide feedbacks on how to modify keywords in Step 2. Rough manual reading can be done by random sampling, or more conveniently from back to front. Since lists of news results by news media databases usually have options to Sort by Relevance, frontlines displayed in the front of the list of searching results are ranked as more relevant to search terms than that of the backlines of the list. ('Sort by Relevance' is one of the sorting functions provided by Lexis Advance, which also provides 'Sort by Date' and 'Sort by Document Title'. Among the three options, 'Sort by Relevance' works best for us to read roughly to change the frequency setting of keywords by trial-and-error. Therefore, 'Sort by Relevance' was chosen before downloading the data from Lexis Advance. Usually, news databases have similar functions for readers to read roughly and conveniently.) A proper percentage, like 80% of results which are relevant among all, can be set to meet our expectation.

To better facilitate future analysis, all downloaded text data will go through structure formatting process. A data structuring program is developed for Lexis Advance to download and organize the text data into structured format. The relevant media articles are processed in order of relevance, and detailed information such as the publication time of the articles, media source, author, article length, etc. are stored in a structured manner. An example of structured media data is shown in Table 4. As for data integration, any news data downloaded from suitable data sources (not only from Lexis Advance) can be arranged and structured in the format of Table 4 through dada cleaning and processing procedure. After data processing, the toolkit provided by this research can be applied to the integrated data regardless of the original data sources of it.

**Table 4.** Example of structured data

| Paper Index | 1 |
|---|---|
| **Title** | The 1997 water rights settlement between the state of Montana and the Chippewa Cree tribe of the Rocky Boy's Reservation: the role of community and of the trustee. |
| **Source** | ASAPII Database |
| **Date** | Dec 22, 1998 |
| **Pg;ISSN;Vol;No** | Pg. 255(1); ISSN: 0733-401X; Vol. 16; No. 2 |
| **Words Count** | 18256 words |
| **Author** | Cosens, Barbara A. |

| Body | I. | INTRODUCTION Established on September 7, 1916 "for Rocky Boy's Band of Chippewas and ... other homeless Indians,"(1) the Rocky Boy's Reservation is home to over 3,000 Tribal members. The Reservation's annual population growth rate is in excess of three percent…(original data is too long for demonstration, here is the excerpt) |

**2.4 Potential Analysis**

The news media dataset of water conflict and cooperation on transboundary rivers allows for varieties of analysis in later stage. This study lists several examples of potential analysis including event extraction, stakeholder analysis, sentiment analysis and topic analysis.

*Event Extraction* from news articles is a conventional application of water conflict and cooperation dataset. Same as what has been achieved by TFDD, water events both conflictive and cooperative, were extracted in relevant political science datasets and news articles (Yoffe & Larson, 2001). Event Extraction requires concise and accurate information recognition and extraction from latent content in text data. Since human coders perform better than machine programming (Howland et al., 2006), human coding event extraction is recommended.

*Stakeholder Analysis* for transboundary rivers is a way to identify who has been involved in transboundary water issues, and the roles they play in the game, i.e., understanding the demands and expectations of the major stakeholders inside and outside the basin, based on typical definition of stakeholder analysis (Smith, 2000). News media represent or reflect the interests of its home country, thus via analysis of news media sources in a transboundary basin, political positions and economic interrelationships between riparian countries and other extra-territorial countries lying outside the basin are uncovered. Longitudinal analysis has capability to depict the trajectories of a stakeholder country's interests and reveal the evolution of stakeholder countries in transboundary water issues.

*Sentiment analysis* on the news media dataset on transboundary rivers can bring the implicit information to the surface (Jiang et al., 2016), since willingness of cooperation and hostility of conflicts often hide behind the news articles. Positive and negative sentiment are closely to dynamics of conflict and cooperation in transboundary water issues, which serve as precursors of significant situational changes. Sentiment lexicons (Khoo & Johnkhan, 2018) or machine learning (Neethu & Rajasree, 2013) are major methods for sentiment analysis in text mining.

*Topic analysis* tells the story about main interests and concerns of the news media, even the stakeholders along with time (Jacobi et al., 2016). Topics concerned along with society development, evolutionary trajectories of transboundary water issues can be uncovered through popular algorithms of topic modelling analysis , such as LDA (Alsumait et al., 2009).

**3 Results**

This section overviews the Global Datasets statistically both in terms of spatial coverage and content coverage, which aims to show the datasets telling stories of conflict and cooperation on transboundary rivers from all aspects in a global scale. To demonstrate the effectiveness of the methodological framework and toolkit, manual reading to check the improvements of data relevance was conducted on four representative basins including Nile, Mekong, GBM, and Columbia.

### 3.1 Overview of the Global Datasets

### 3.1.1 Spatial Coverage

### *(1) Continental Coverage*

With the customized search strings for each transboundary river and the data structured program developed for Lexis Advance to organize the data, as of March 10, 2019, the data volume results of 286 transboundary river basins around the world are shown Figure 3 - Spatial Coverage. In Figure 3, the base map of transboundary river basins around the world was downloaded from TFDD in the format of GIS shapefiles (Transboundary Freshwater Dispute Database, 2008).

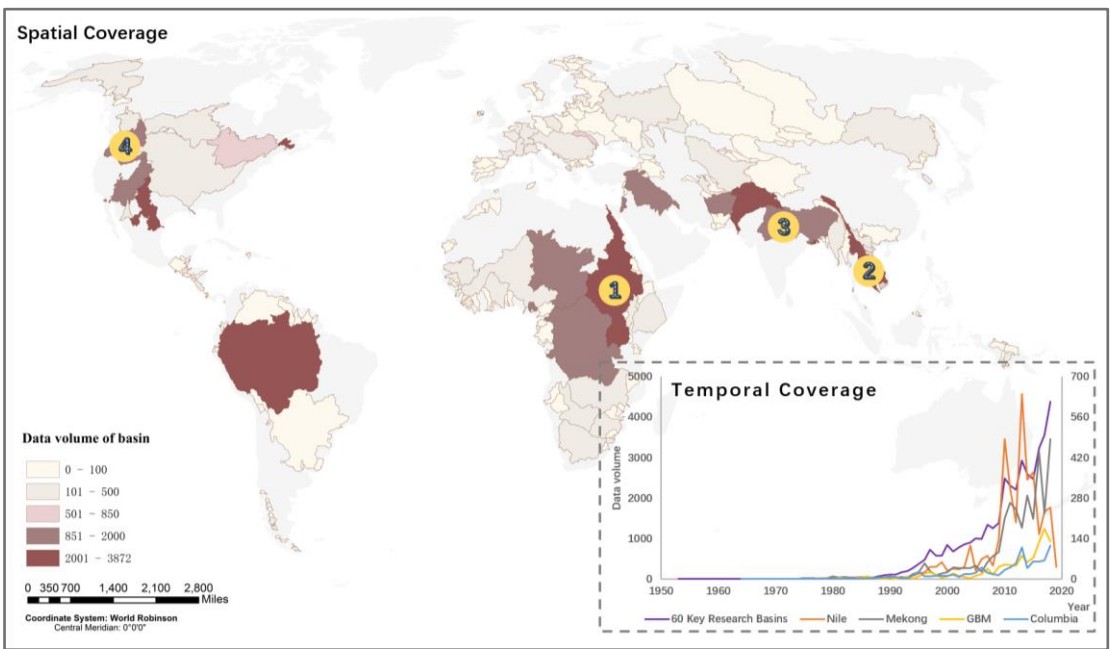

**Figure 3.** Spatial coverage and temporal coverage in basin scale (①Nile; ②Mekong; ③GBM; ④Columbia)

Data volume of news articles reflects the prominence of the conflict and cooperation events discussed in transboundary river basins. Enough data volume promises statistical significance. The mainstream application of this news media dataset is further text mining to track conflict and cooperation dynamics on transboundary rivers. For text mining purpose, this study assumes arbitrarily that 100 media articles are the minimum data volume to track dynamics transboundary rivers along with time.

Overall, there are 60 river basins with more than 100 media articles, which are considered as the Key Research Basins of transboundary water conflict and cooperation in our research. The number of news articles discussing these 60 Key Research Basins reached more than 41,000. Among the 60 Key Research Basins, 16 river basins have more than 850 data records as shown in Table 5, which attract more attention and are considered as Heated Basins. Note that the definition criteria of Key Research Basins (more than 100 articles) and Heated Basins (more than 850 articles) are flexible and adaptive according to specific research demands.

**Table 5.** 16 Most-discussed Basins with more than 850 records

| Order | Basin Name | Continent | Number of records | Countries |
|---|---|---|---|---|
| 1 | Nile | Africa | 3872 | Burundi, Central African Republic, Egypt, Hala'ib Triangle, Eritrea, Ethiopia, Kenya, Rwanda, Sudan, Abyei, South Sudan, United Republic of Tanzania, Uganda, Dem. Republic of the Congo |
| 2 | Mekong | Asia | 3253 | China, Cambodia, Lao People's Democratic Republic, Myanmar, Thailand, Viet Nam |
| 3 | Rio Grande (N. America) | North America | 2718 | Mexico, United States of America |
| 4 | Indus | Asia | 2404 | Afghanistan, China, India, Nepal, Pakistan |
| 5 | St. John (North America) | North America | 2356 | Canada, United States of America |
| 6 | Amazon | South America | 2078 | Bolivia, Brazil, Colombia, Ecuador, French Guiana, Guyana, Peru, Suriname, Venezuela |
| 7 | Colorado | North America | 1975 | Mexico, United States of America |
| 8 | Jordan | Asia | 1816 | Egypt, Israel, Jordan, Lebanon, West Bank, Syrian Arab Republic |
| 9 | Congo/Zaire | Africa | 1391 | Angola, Burundi, Central African Republic, Cameroon, Congo, Gabon, Malawi, Rwanda, Sudan, South Sudan, United Republic of Tanzania, Uganda, Dem. Republic of the Congo, Zambia |
| 10 | Lake Chad | Africa | 1353 | Central African Republic, Cameroon, Algeria, Libya, Niger, Nigeria, Sudan, Chad |
| 11 | Ganges-Brahmaputra - Meghna | Asia | 1183 | Bangladesh, Bhutan, China, India, Myanmar, Nepal |
| 12 | Helmand | Asia | 1168 | Afghanistan, Iran (Islamic Rep of), Pakistan |
| 13 | Cross | Africa | 1110 | Cameroon, Nigeria |
| 14 | Tigris-Euphrates/Shatt al Arab | Asia | 939 | Iran (Islamic Rep. of), Iraq, Jordan, Saudi Arabia, Syrian Arab Rep., Turkey |
| 15 | Columbia | North America | 859 | Canada, United States of America |
| 16 | Tijuana | North America | 853 | Mexico, United States of America |

Most studies of conflict and cooperation on transboundary rivers focus on individual basins, which seeks solutions to dealing with local challenges on transboundary water resources (Bernauer & Böhmelt, 2020). Therefore, formation of general understanding of conflict and cooperation on transboundary rivers needs global data support besides expert on-site experience from research of individual basins. Many most-discussed transboundary river basins such as the Nile, Mekong, Indus, GBM, and Tigris-Euphrates/Shatt al Arab etc. are located in regions featured with frequent tensions and armed conflicts (Pohl et al., 2014), and are well-known by people. However, this study finds that there are also some river basins from the authors' point of view, which less attention has been paid to in the past in terms of transboundary water conflict and cooperation research, e.g., St. John River (North America), and Tijuana River.

Data volume of transboundary water conflicts and cooperation news articles in the datasets of 60 Key Research Basins on
different continents: for Asia is 14454, for North America is 11306, for Africa is 10734, for Europe is 2674, for South America
is 2498. It could possibly be attributed to the discrepant levels of economic development of major countries on each continent,
or varied attention paid to discussion of management of transboundary rivers. The other important reason could be the linguistic
variations. Since this paper chose English newspaper as the search scope, the large amount of data in North America and the
small amount of data in Europe could be due to system bias caused by language preferences.
There are notably large amount of transboundary water conflicts and cooperation events reported in Asia and Africa, which
indicates that transboundary water management is a major topic of peace and development in both Asia and Africa. Taking
into consideration that most countries on these two continents do not speak English as their mother language, the existence of
a large number of news media articles on transboundary water conflicts and cooperation between Asia and Africa, on the one
hand, reflects the fervent concerns about the transboundary water resources, and the desires for peace and development; on the
other hand, it also reflects that people around the world are more involved in transboundary water issues in Asia and Africa,
and have invested heavily in the development and construction and pay close attention to these two rapidly developing and
eye-taking continents.
*(2) National Coverage*
News media data volumes in the datasets of 60 Key Research Basins from different countries in the world are shown in Figure
4 - Spatial Coverage. In Figure 4, the base map of countries around the world was downloaded from ArcGIS Hub in the format
of GIS shapefiles (Esri Data and Maps, 2021). It is seen that United States of America contributes 11515 news articles on
transboundary water conflict and cooperation, ranking number one, both as a riparian stakeholder in the transboundary water
issues with Canada and Mexico, and as an extra-terrestrial international stakeholder involving in the transboundary water
issues on continents other than the North America. Since a country's development and utilization of transboundary freshwater
resources inevitably involves relations with other riparian countries, and transboundary water cooperation and conflicts often
involve broader economic and social ties between riparian countries, transboundary freshwater management is an important
component of the diplomacy of riparian countries; on the other hand, due to factors such as global hegemony, transnational
investment, colonial history and other factors, transboundary freshwater management often involves countries outside the
region, becoming a stage for great powers to play (Mirumachi, 2015).

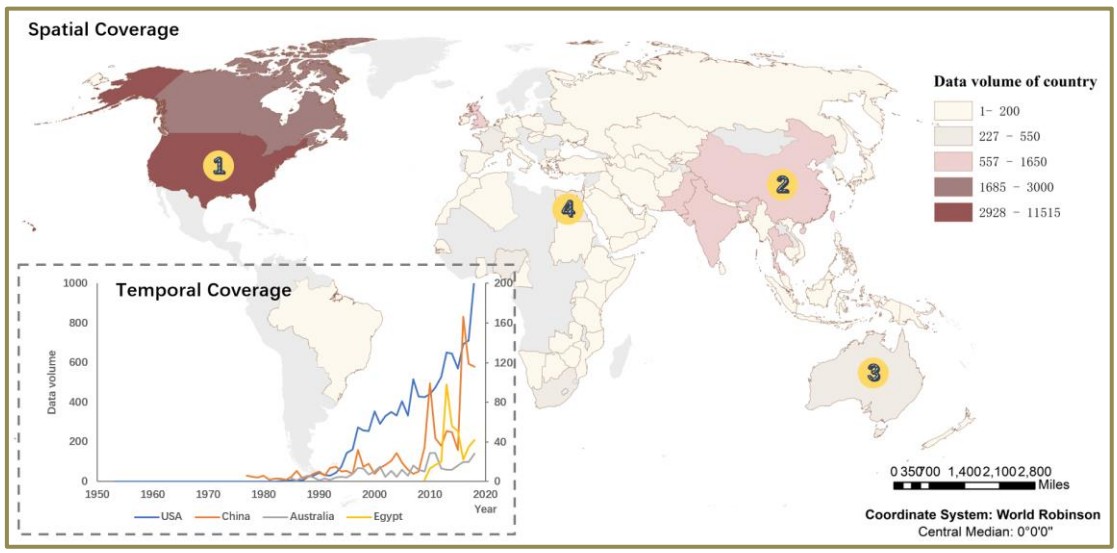

**Figure 4.** Spatial coverage and temporal coverage in country scale (①USA; ②China; ③Australia; ④Egypt)

**3.1.2 Temporal Coverage**

Temporal coverage of the datasets of 60 Key Research Basins (stated in Spatial Coverage section) and four case study basins are shown in Figure 3 - Temporal Coverage, which shows how many news media articles released along with years on transboundary water conflict and cooperation. Noted that due to differences of order of magnitudes, data series of 60 Key Research Basins uses the major vertical axis which ranges from 0 to 4500, and the four case study basins share the minor vertical axis which ranges from 0 to 700. The datasets cover from the year of 1953 to 2019. Boom of news articles on transboundary water conflict and cooperation emerges from 1990s, and potentially continues in the future. That emphasizes the necessity to revise the methodological framework and toolkit for news media dataset tracking of conflict and cooperation dynamics on transboundary rivers to cope with the era of big data. For the four case study basins, the changing trends of data volume display strong vibrates, which may be affected by certain water events and geopolitical relations in the river basins at the moment.

Temporal coverage of four representative countries, which are United States of America (USA, using the vertical minor axis on the left), China, Australia and Egypt (using the vertical major axis on the right), is shown in Figure 4 – Temporal Coverage. USA contributes the largest volume of data among countries in the world; China promotes transboundary cooperation in Mekong River Basin actively in the recent years; Australia does not have a transboundary river with other neighbouring countries, but releases lots of news articles on transboundary water issues; and Egypt is one of the major countries in Nile River Basin, which is representative in transboundary water conflict and cooperation. Same with the temporal coverage of basin analysis, country datasets also cover from the year of 1953 to 2019. Data volume took off from 1990s, and potentially continues in the future as well. For the four representative countries, the overall trends of data volume go up along with time and are affected by contextual events in the country to show strong vibrates.

### 3.1.3 Content Coverage

Word frequency analysis demonstrates that this study has generated good datasets tracking of conflict and cooperation dynamics on transboundary rivers. In the datasets, words concerning with water body function, hydraulic infrastructure construction, basin management, national power, civic rights, jointed research and water conflict and cooperation appear in a high frequency, consistent with the related keywords in TFDD (Yoffe & Larson, 2001) and relevant words provided in UNBIS Thesaurus (UNBIS Thesaurus, 2021). This indicates that the datasets are closely corresponding to the research question, providing data as needed.

### 3.2 Relevance Screening

The major advancement of this methodological framework is that it allows efficient and effective tracking of transboundary rivers conflict and cooperation events. The keywords generator developed in this study could result in an acceptable level of relevance without too much manual coding intervention. To demonstrate the effectiveness of this methodological framework, four river basins: Mekong, Columbia, Nile, and GBM were taken as case studies to conduct manual coding process. Two manual coders, who were trained beforehand, were involved to work independently for the four basins in the coding process. Each one undertook half of the total workload in which articles in the datasets were divided into two groups randomly. Before starting, inter-coder reliability test was conducted. The test randomly selects 50 articles from the datasets for two coders to read, differences were then discussed, and definitions were given to reach common understanding. Krippendorff's Alpha-Reliability was calculated as 0.81, which is considered as valid and consistent (Krippendorff, 2004).

The total number of downloaded articles, after removing duplicates by the function of removing duplicates in the data panel of Microsoft Excel, and the remaining number of relevant articles with removal of the duplicates are shown in the Table 6. The calculation Equation of Relevance Percentage is shown as Eq.1.

**Table 6.** Manual reading results of representative river basins

| Basin name | Number of downloaded | Number after removing duplicates | Number after removing irrelevant | Relevance percentage(%) |
|---|---|---|---|---|
| Nile | 3872 | 3563 | 3164 | 88.80 |
| Mekong | 3253 | 2917 | 2291 | 78.54 |
| GBM | 1183 | 1092 | 724 | 66.30 |
| Columbia | 859 | 817 | 295 | 36.11 |

$$\text{Relevance percentage} = \frac{\text{Number after removing irrelevant}}{\text{Number after removing duplicates}} \times 100\% , \tag{1}$$

The last column of Table 6 shows the Relevance Percentage for the four river basins in a descending order. The relevance percentage of Nile, Mekong and GBM are at acceptable level, and that of Columbia is less satisfying. This is due to Columbia belonging to special basin name category, details shown in 2.2.2 a), whose basin name is same as the name of a certain riparian country or state. To further investigate of relevance percentage of the four basins, the relevance percentage in 10% stepwise is

calculated for each basin using Eq. 2. The relevance percentage in 10% stepwise for the four basins is shown in Figure 5. In
Figure 5, the horizontal axis is every 10% Stepwise segment of the news media articles data, and the vertical axis indicates the
Relevance Percentage of that segment of data.
Relevance percentage in 10% stepwise  =
Relevance percentage for every 10% of the total number after removing duplicates ,                            (2)

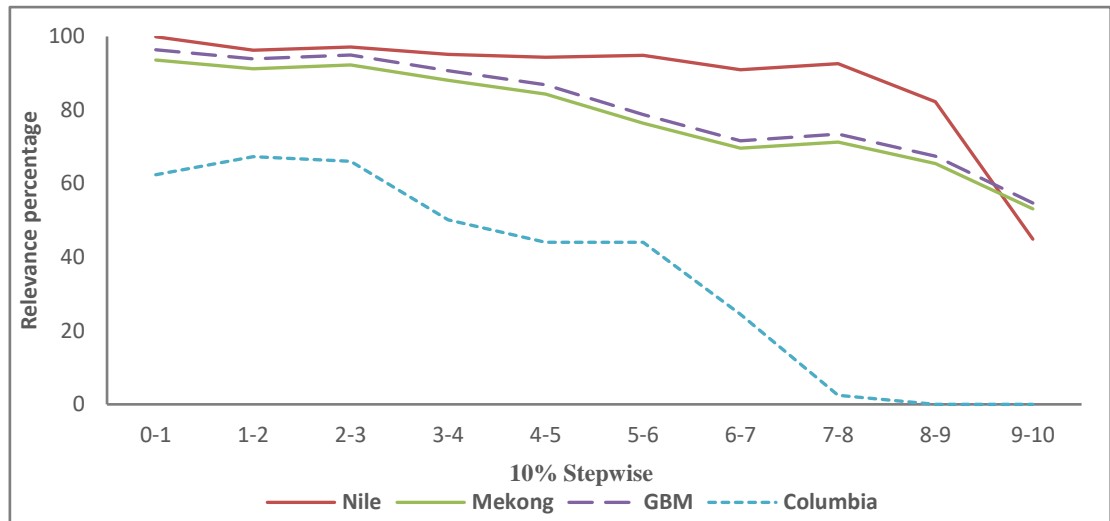


**Figure 5.** Relevance Percentage in 10% Stepwise for the four basins

The purpose of Figure 5 is to demonstrate the necessity to apply special treatments for some river basins. Since the datasets
retrieved from Lexis Advance are sorted by relevance, frontlines are naturally more relevant than the backlines, and the
Relevance Percentage in 10% Stepwise displays descending trendlines. However, slopes of the trendlines of relevance
percentage in 10% stepwise between basins reflect heterogeneity of data quality. The Relevance Percentage for Columbia is
unsatisfactory even in the first 10% of the article list, since 'Columbia' is both a district's name and a commercial brand, listed
in Table 1. It makes sense that the data quality of Columbia River Basin is not as good as others. Special treatment for Columbia
should be adopted here to improve the data quality, as well as special treatments are needed for certain categories of basins
and corresponding treatments as mentioned in Table 1. To do so, usually enforcement of the frequency constraints shown in
Sect. 2.2.3 (i.e., raise the frequency setting for 'water' and 'river' to filter out the geopolitical articles as many), or removal of
the most irrelevant articles in the end of the dataset work well. With an anticipation of relevance percentage in mind, random
sampling or manual reading of the last percentage of articles are often undertook to check the data quality. For example, given
the Relevance Percentage in 10% Stepwise for Columbia, raising the frequency setting of 'water' and 'river' to 5 times, or
removal of the last 40% of the data retrieved in the Original Dataset due to its low relevance in general are feasible solutions
to improve data relevance. For other basins with satisfactory data relevance, no further operation is needed; and for the other
basins, similar operations as for Columbia River Basins can be adopted before further potential analysis.

**4 Summary**

Management of transboundary rivers is challenging both in terms of political and environmental in the 21st century. Data
support is crucial for research of conflict and cooperation on transboundary rivers. Conventional construction manner of dataset
by manual reading and extraction cannot meet the requirement for fast-updating in the big data era. This study brings up a
revised methodological framework based on the conventional and toolkit for news media dataset tracking of conflict and
cooperation dynamics on transboundary rivers. Design of the framework follows closely Lasswell' communication model
(Lasswell, 1948) involved with seven elements- "who, with what intentions, in what situations, with what assets, using what
strategies, reaches what audiences, with what result". Basic search keywords were adopted from TFDD and further revised to
include five blocks of terms to make it extensible and adjustable according to a certain research topic. Through Block 1 and
Block 2 with corresponding toolkit (shown in Figure 2), a dataset covering transboundary rivers in a global scale can be
generated, which is improved than results of TFDD. All the special treatments for basin names (shown in Table 1), country
names (shown in Block2), and term frequency setting of keywords (stated in Section 2.2.3) are crucial measures to enhance
data quality and save manual efforts, which are improvement beyond achievements of TFDD. Following the methodological
framework, a dataset with good trade-offs between data relevance and coverage is generated. This study demonstrates the
effectiveness of the framework and the potency of our toolkit. This framework possesses extensibility and compatibility to
other research topics besides transboundary water resources management since the search terms are adaptive and the toolkit is
transplantable for related future research. With this revised framework and toolkit, research using news media tracking of
conflict and cooperation dynamics on transboundary rivers will be much easier and more practicable.
Implications of this research can be manifested through how we can use the news media dataset generated by the
methodological framework and toolkit tracking of conflict and cooperation dynamics on transboundary rivers. The dataset can
serve as the foundation for further analysis, e.g., to study the attitudes, the topics of concerns, and the relationship between the
evolution of the water governance network and the level of integrated water management along the evolution of water conflict
and cooperation in the transboundary river basins. Ultimately it can contribute to understanding of the driving mechanisms
and transformation laws of water conflict and cooperation. By capturing the characteristics of life cycles of water conflicts and
cooperation, future researchers can explore the temporal evolution trend and spatial distribution law of global transboundary
water conflicts and cooperation events, as well as the guiding significance of appropriate policy intervention, and improve the
level of global water security.
Meanwhile, this research and the dataset can also serve as a methodological and statistical foundation of quantifying the social
dimension in socio-hydrological approaches of understanding transboundary river system. Recent attempt has been made to
take socio-hydrological approach to tackle the feedback mechanism of co-evolved sub-systems (Lu et al., 2021). While socio-

hydrological model can contribute to understanding the complexity of the intertwined nature of transboundary river system, quantifying the social variable has been challenging in general. There has been increasing recognition that news media provides a valid proxy to reflect the changing values and interest of each riparian country (Wei et al., 2021), conflict and cooperation sentiments that reflected from news article have been adopted in socio-hydrological models as the willingness of cooperation to validate the social sector of the model (Lu et al., 2021). When expanded to other river basins, this study could provide a methodological support in measuring the social sector of transboundary river systems more effectively.

Still this study has some limitations which could be overcome in following researches: (1) absence of newly-registered rivers: the list of transboundary rivers adopted in this study includes 286 rivers, which could be expanded to 310 rivers in the near future; (2) language limitation: the scope of this study limits to English newspaper only due to our limitation of language processing, which could be expanded to include more main languages and local languages in transboundary river basins; (3) absence of tributary information: in the keywords generator, tributaries of transboundary rivers are not included, which may lose content coverage to some extent. Future work can add more details concerning tributaries of transboundary rivers.

*Code and data availability.* The data and code used in this study are publicly available on Zenodo (including: basin-country dictionary; dictionary of country names with different formats (special country dictionary); dictionary of basin names with different formats; python code of searching term generator. DOI:10.5281/zenodo.5112624

*Author contributions.* LG, JW, and FT designed the research framework. LG collected data and conducted data analysis. LG, JW, KZ, and J.L.W conducted manual reading for the case studies. LG, JW and FT composed the manuscript with contribution from KZ and J.L.W.

*Competing interests.* The authors declare that they have no conflict of interest.

*Special issue statement.* This article is part of the special issue "Socio-hydrology and transboundary rivers". It is not associated with a conference.

*Acknowledgements.* We would like to acknowledge the National Key Research and Development Programme of China (grant no. 2016YFA0601603) for the funding and support of this research.

*Financial support.* This research has been funded by the National Key Research and Development Programme of China (grant no. 2016YFA0601603).

*Review statement.*

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

501

502

## Appendix

Recordings of trial-and-error process are provided as follows to demonstrate the effects of various groups of frequency settings of keywords and how balance between relevance and coverage is approaching. Two justification indicators of data relevance are adopted: (1) Indicator 1: the number of articles relevant to our research topic within 20 articles at 60% of total data volume. For example, there are 10000 articles retrieved by the certain frequency setting of search terms in Lexis Advance, we locate the article at exactly 60% of total data volume, which is the $6000^{th}$ articles, and read 20 articles from there. Therefore, Indictor 1 is how many articles are relevant among $6001^{st}$-$6020^{th}$ articles. (2) Indicator 2: the number of articles relevant to our research topic within 20 articles at 80% of total data volume. Similar to the algorithm of Indicator 1, if the total data volume is 10000, Indictor 2 is how many articles are relevant among $8001^{st}$-$8020^{th}$ articles. Table 7 presents the results of trial-and-error process of frequency settings of keywords for Mekong, Nile and Jordan River Basin, and shows that strong frequency settings enhance data relevance prominently, and at the same time reduces data volume to a large extent. To promise a balance between data relevance and coverage, proper frequency settings of search keywords should be adopted. In this study, Test 6 is adopted as the final setting. Notice that Nile and Jordan River Basin have overwhelmingly large volume of data if no additional constraint are exerted, therefore ("Nile River" OR "Nile Basin" OR "Nile Water") or ("Jordan River" OR "Jordan basin" OR "Jordan water") are added to basic search terms to limit data volume to an acceptable extent. While conducting trial-and-error processes, topics of irrelevant articles are also recorded to show the potential causes of irrelevance and may provide us some hints to modify the search terms for a better performance, shown in Table 8.

**Table 7.** Trial-and-error process of frequency settings of keywords for Mekong, Nile and Jordan

| Test index | Frequency settings | | | | Mekong | | | Nile | | | Jordan | | |
|---|---|---|---|---|---|---|---|---|---|---|---|---|---|
| | Basin name | Riparian country | Water | River | Data volume | Indicator 1 | Indicator 2 | Data volume | Indicator 1 | Indicator 2 | Data volume | Indicator 1 | Indicator 2 |
| 1 | 1 | 1 | 1 | 1 | 27975 | 5 | 2 | 16227 | 8 | 4 | 28604 | 3 | 0 |
| 2 | 3 | 1 | 1 | 1 | 7536 | 13 | 10 | 6707 | 17 | 15 | 14028 | 4 | 1 |
| 3 | 5 | 1 | 1 | 1 | 4036 | 16 | 15 | 4157 | 19 | 16 | 13284 | 7 | 3 |
| 4 | 5 | 2 | 1 | 1 | 3695 | 16 | 16 | 4124 | 20 | 16 | 13263 | 7 | 1 |
| 5 | 5 | 2 | 2 | 2 | 3316 | 18 | 17 | 4017 | 20 | 18 | 5267 | 5 | 3 |
| 6 | 5 | 2 | 3 | 3 | 3102 | 19 | 17 | 3912 | 20 | 18 | 3830 | 7 | 10 |

**Table 8.** Recordings of potential irrelevant topics for Mekong, Nile and Jordan

| Mekong Test Index | Data volume | Indicator 1 | Indicator 2 |
|---|---|---|---|
| 1 | 27979 | Paper index: 16787-16806 | Paper index: 22383-22402 |
| | | 16789: Coastal monument;16790: Catfish;16791: Missing American servicemen;16792: Missing people;16793: Business plan;16795: Life-style;16796: America navy river corps;16798: Life-style;16799: Grade nationale;16800: Travel to Vietnam and Cambodia;16801: Travel;16802: Travel along the river;16803: Travel;16804: Riots in Thailand;16805: Riots in Thailand; | 22383:Family;22384:Soldiers;22385:Vietnam annexed Cambodia through reconciliation;22387:Vietnam sentences followers after trial of Buddha;22388:Culture;22389:Judicial delays in four cases;22390:Ban on swill feed;22391:Combine harvester race;22392:Relocate 7 million people to relieve pressure on overcrowded areas;22393:The man who reformed the United States Navy;22394:Books about Vietnam;22395:China plans to establish a national park system;22397:Image consulting;22398:Songkran Festival of Thailand;22399:Southeast Asian refugees;22400:Bow movement;22401:A dueling event in Denver;22402:Cambodian adoptees; |
| 2 | 7536 | Paper index: 4521-4540 | Paper index: 6029-6048 |
| | | 4522:Former Vietcong say fighting on the Mekong River;4524: Elephant;4528: Luang Prabang;4533: Laotian culture and beauty;4534: Mekong River Journey;4537: Hero for Children's Rights;4540: Travel to Cambodia; | 6029:Crossing the Mekong;6030-6031:Vietnam's rice exports;6032:DNA catch;6033:Vietnam cruise;6041:Inland river cruise;6043:National Geographic researcher Reno;6044:Mekong River Tourism;6046:Escape the molecular;6048:Buddha; |
| 3 | 4036 | Paper index: 2422-2441 | Paper index: 3229-3248 |

| | | | |
|---|---|---|---|
| | | 2428:A cruise ship on the Mekong River;2434:Mekong River travel;2440:Illegal timber trade;2441:Mekong River travel; | 3231:Mekong Animals;3235:A manhunt for missing American soldiers in Laos;3236:First impressions of Cambodia and Vietnam;3239:Travel to Vietnam and Cambodia;3240:Illegal logs are cut down; |
| 4 | 3695 | Paper index: 2217-2236 | Paper index: 2956-2975 |
| | | 2218:Travel to Vietnam and Cambodia;2220:Travel to Laos;2231:Pacific Command disaster response exercise in Vietnam;2236:Vientiane, capital of Laos; | 2962:Visit Southeast Asia;2963:Cambodia is trying to save a rare Mekong river dolphin;2968:A search for missing American soldiers in Laos;2975:Travel to the Mekong; |
| 5 | 3316 | Paper index: 1990-2009 | Paper index: 2653-2672 |
| | | 1997:Mekong Tourism;2007:Mekong Prize winner | 2662:Mekong River travels in Thailand, Cambodia and Vietnam;2663:Travel to Thailand;2671:Mekong navigation; |
| 6 | 3102 | Paper index: 1860-1879 | Paper index: 2482-2503 |
| | | 1861:Mekong Adventure; | 2483:Laos arrests American manhunt for missing man;2486:Roaming along the Mekong river;2488:Drifting; |

522

| Nile Test Index | Data volume | Indicator 1 | Indicator 2 |
|---|---|---|---|
| 1 | 16227 | Paper index: 9736-9755

9736-9738:Rebels in South Sudan;9739-9740:Archaeologist;9741:Israel may withdraw from the West Bank;9744:Slavery in ancient Egypt;9745:Kurdish rebels in Turkey;9748:Travel to Egypt;9750-9752:Egypt's interior minister refused to allow "militias" to enter; | Paper index: 12982-13001

12982:Farmers and the Egyptian government fought for years in a legal battle;12983:Curseja Island;12984:Detectives use modern science to solve a 3,300-year-old murder mystery;12985:Violence in Egypt;12987:Tagore festival in Egypt;12988:Bossi language learning courses;12989-12992:Russian plane crash;12993:Egypt's population grows;12995:Reviving Egypt's tourism industry;12996-12997:Anti-government demonstrations in Sudan;12998:Tourism landscape;13001:Egyptian court holds second mass trial; |
| 2 | 6707 | Paper index: 4024-4043

4037:Lake Victoria renamed;4038:Egyptian journalist Resigns;4041:Travel along the Nile; | Paper index: 5366-5385

5371-5373:Luxor Temple;5378:Egyptian history;5380:The uprising in Egypt is resurgent; |
| 3 | 4157 | Paper index: 2494-2513

2509:Changes along the Nile; | Paper index: 3326-3345

3326:Conflict in South Sudan;3328:Travel along the Nile;3333:Sudanese refugees;3343:Sudan Peace Conference; |
| 4 | 4124 | Paper index: 2474-2493

No irrelevant articles | Paper index: 3299-3318

3301:Egypt will withdraw 3.4 million from federal reserves to meet food needs;3302:Nile culture;3305:Sudan earthquake sequence 1990-1991 and the extent of the East African Rift Valley system;3317:Ethiopia: Lakeside cities overcome Africa's tourism crisis; |
| 5 | 4017 | Paper index: 2410-2429

No irrelevant articles | Paper index: 3214-3233

3216:Displaced South Sudanese;3232:Sudan earthquake sequence 1990-1991 and the extent of the East African Rift Valley system; |
| 6 | 3912 | Paper index: 2347-2366

No irrelevant articles | Paper index: 3130-3149

3134:An English man crosses the Nile on foot;3136:Displaced South Sudanese; |

523

| Jordan Test Index | Data volume | Indicator 1 | Indicator 2 |
|---|---|---|---|
| 1 | 28604 | Paper index: 17162-17181

17162: Gaza's prison;17164: Israel security Separation Wall;17165: Jesus through Anne Rice's eyes: A book review;17166: The king of Morocco visited the United States;17167: Jewish terrorist group;17168:Palestinians demonstrated in israeli-occupied territory;17169:Jordan's king urged the Palestine Liberation Organization to recognize Israel;17170-17172:United States: Israeli-Palestinian peace agreement;17173: Riding;17174:The PLO will not meet with American officials;17175:There has been violence in Jerusalem;17176:Israel's democracy and Arab population;17177: Women go to Palestine to resolve violence;17178: Music;17181: Hotel prices in Australia have fallen along with Asian growth; | Paper index: 22883-22902

22883:Joshua Myron, zionist who fought the Turks, died;22884: Did Netanyahu explain why the Palestinians did not reach a deal;22885: Ottawa ordered compensation for disabled first Nations children;22886: Sacramento State University student arrested in terrorist ring;22887: South Africa: Two white-owned farms to be confiscated in land reform;22888: State Department envoy meets with Palestinian Christians who oppose Israel;22889: Three-year-old girl shot dead in Gaza;22890:Mormon Temple renovation;22891: Yahsat hosted a forum on the humanitarian use of satellite broadband;22892: Humanitarian work;22893: Jenkins' death;22894: Interfaith activity;22895: The ultimate consultant;22896: Exhibition in the west bank;22897: The Jewish population in the occupied West Bank is set to more than double this year;22898:Police have arrested a suspect in the Maverick shooting;22899:Disturbing violence;22900: "The etymology of the names Israel and Jacob;22901: Terrorism;22902: The city of Midville's first citywide master plan for trails; |
| 2 | 14028 | Paper index: 8417-8436

8417: Missionary trip to the West Bank;8418: Against Islamic State militants;8419: Israel imposed sanctions on Gaza;8420: Arafat;8421: Israeli withdrawal;8422: The Israeli military has questioned an Arab mayor in the West Bank;8423:It is widely believed in Israel that the current situation in Judea, Samaria and Gaza cannot and should not continue;8424: Arafat rose up;8425: Palestinians say Washington | Paper index: 11222-11241

11222: Manitoba: The Assembly of First Nations supports the Declaration of the Manitoba Chiefs;11223: The Sheikh Hussein Bridge across the Jordan River was completed;11224: The Israelis shot at two Jordanians;11225: The sick Menachem Begison;11226: Top election official supports south Jordan petition;11227: How did a high school student in Nuremberg talk about crossing Israel;11229-11230: Protesters in Amman burn an Israeli flag after the judge's killing;11231: SCR 591 recognizes Sao Paulo's historic landmarks and museums;11232: Expand light rail public transport; |

| | | | |
|---|---|---|---|
| | | accepts their approach to ending attacks on Israel;8426:The Likud trounced Sharon;8429:Pope endorses' Palestinian Aspirations';8430:The Israeli authorities have jailed three senior Palestinian leaders without trial;8433:Defense of the Jewish State;8434:Australia's relationship with Israel;8435:Former commander of the Arab Legion Grubb Pasha has died;8436:A leadership void is holding Egypt back; | 11233:Update from AFPTV on Tuesday;11234-11236:Palestinians in the West Bank are under increasing economic pressure;11237-11239:Sharon army;11240:Silt diversion walls in East Jordan;11241:Traveler's cheque; |
| 3 | 13284 | Paper index: 7970-7989<br><br>7970:Israel faces a demographic threat;7971:Israel and Palestine live side by side in peace;7972:Peace negotiation;7973:Palestinian Elections postponed;7974:The countryside camping;7975:A week news;7976:The American president meets with Jordan's king;7977:Watch the Pope's Middle East pilgrimage online;7979:A secret meeting of Arab and Israeli writers;7983:Top story on Tuesday;7984-7985:Better support for Aboriginal children;7986:Israel Archives;7963:Catholics and Muslims seek dialogue; | Paper index: 10627-10646<br><br>10627-10629:Individual account;10630:Witness: The Jordanian defendant had ties to Osama bin Laden;10631:The wounded mayor vowed to continue the fight for Palestinian rights;10632:Former U.S. PRESIDENT: Middle Eastern leaders must tell their people that compromise is honorable;10633:Jordan baptism site sells bottled holy water tender;10637:A letter from Israel;10638:Public money spent on Park Avenue;10639:The Israeli prime minister has proposed the creation of an independent Palestinian state;10640:The European Commission has issued a final warning to The UK over repeated violations;10641:Vote split;10642-10643:Jordan's parliament failed to overthrow the government;10644:Jordan Valley Trail;10645:Possible hazards caused by pumping water near rivers;10646:Task biography; |
| 4 | 13263 | Paper index: 7958-7977<br><br>7958-7959:Terrorism;7960:The new chief rabbi is a firebrand nationalist;7962:Silt diversion wall;7963:Catholics and Muslims seek dialogue;7964:The Palestinian government's plan;7965:Palestinian refugees;7966:Jordan bridge;7968:The United States is pushing for a Middle East peace plan;7969:The Arabs conspired to blockade Israel;7970:Jordan River Cycle Path;7971:Mr. Netanyahu linked peace to Palestinian recognition of Israel as a Jewish state;7976:Jordan refugee woman craftsman; | Paper index: 10610-10629<br><br>10610-10611:Jordanian-british student volunteers seek positive change in Western and Arab societies;10612:Jerusalem Liberation Army;10613-10614:Palestinian textbooks versus Israeli textbooks;10615:Some rare right whales like winter in Maine;10616:The PLO is preparing to move to Gaza and Jericho;10617-10618:Arab Bank employees volunteer in Aguilon;10619:East Jordan Commissioner Thomas Breney asked the board to vote on hiring a full-time fire chief;10621:The FBI is demanding payment for the local youth's treatment;10622:The federal government is seeking to appeal the ruling on medical costs;10623-10624:Negotiations between Israel and Egypt;10625:The Conservative Party is appealing against the treatment ruling;10626:Fatah's Al-Aqsa Brigades killed a woman in Nablus accused of collaborating;10627:Mr Hotovili bemoans Likud "schizophrenia" over two countries;10628:Kiryat Arba population;10629:In the Likud debate, the two-state solution is schizophrenic; |
| 5 | 5267 | Paper index: 3160-3179<br><br>3160:Mr Netanyahu does not fear being blamed if the London meeting is inconclusive;3161:Forty-eight hours in Amman;3162:Israel has begun clearing land mines at the site of Jesus' baptism in the West Bank;3164:Covenant of Israel;3166:Israel's efforts to win Over Christian tourists;3168:Christian sites are covered in landmines;3170:Community Notes - Volunteer;3171:Edit the letter in the pouch;3173:Travel;3174:Mormons and non-Mormons;3175:Immigrants find peace and opportunity in Corona;3176-3177:Hymns to Haaznu on a biblical urn;3178:A joyous gathering of prominent Israeli and PLO officials in the three years since the Signing of the Oslo Accords;3179:The new chief rabbi is a firebrand nationalist; | Paper index: 4214-4233<br><br>4214:Jordan sewage pump;4215:Across the border;4216:In memory of a distinguished journalist;4217:Travel manuscript;4218:Some news;4219:Joint Technology Center;4220:leprosy;4221:Israeli soldiers pass through barbed wire in Wazzani;4223:Storm hits northern Michigan;4224:Jon and Martha Jensen of Petoskey gave birth to daughter Ruby Susan at Northern Michigan Hospital;4225:Peter and John stood before the Sanhedrin;4228:Obituary;4229:The UAE supports Jordan in implementing its development plans;4230:Russia and the United States are set to reach a new agreement by next summer on deep cuts in strategic offensive weapons;4231-4232:Hussein riots;4233:The East Jordan Chamber of Commerce distributes community awards; |
| 6 | 3830 | Paper index: 2298-2317<br><br>2298:The establishment of a provisional Palestinian state;2299:Ways to promote Jordanian and British military cooperation;2301:Israeli Prime Minister Benjamin Netanyahu has said he will see the final results of a peace deal;2303:Epiphany;2306:Winners of the first Tourism Promotion Peace Prize have been announced;2308-2309:Better support for Aboriginal children;2310:Jordan has "ridiculed" Israeli ministers' efforts to block Palestinian statehood;2311:Pope Francis is making a visit to the Holy Land;2314:The Jordan River has long been a source of entertainment for wasatch central city dwellers;2315:Community news;2316:South Jordan celebrates its 150th anniversary;2317:Jordan Trail; | Paper index: 3064-3083<br><br>3064:American troops were sent from California to Utah during the Civil War;3065:A Seattle moving company is offering spring deals;3066-3067:Seattle Moving Company;3068:Opponents of sewage treatment plants;3069:Soldiers swept away by the Jordan River;3073:Civil servants are considering a one-day strike on Monday;3074:Pilgrims mark baptism traditions in the Jordan River;3075:Word games;3076:People's Fund grant project; |

524