# Peer review of "Building a methodological framework and toolkit for news media"

_Hydrology and Earth System Sciences, 2021_

## Author Comment (AC1)

**Authors' Response to Referees' Comments**

We would like to thank both Reviewers for their constructive comments, which will help improve the manuscript substantially. In the following, we provide answers to all the comments on a point-to-point basis. Each answer is structured as follows: (1) **RC#** comments from Referees, (2) **AR#** author's response.

**Authors' replies to Comments of Reviewer #1:**

**RC#1:** This paper develops a toolkit to track media data on conflict and cooperation events in transboundary river basins. It is a praising approach to provide systematic collection and mining of mass qualitative data in transboundary river basin management. The paper provided detailed procedures on how such toolkit should be implemented, and further illustrated its applicability with various case rivers. It was presented in a well-structured manner, however, there are some concerns that needed to be addressed.

**Authors Response#1:** We would like to thank reviewer #1 for the positive comments, which we believe will help to improve the manuscript substantially. We will rework on the main issues pointed out by the reviewer to progress the manuscript further. Our explanations and responses to all the reviewer's comments and questions are listed below.

**RC#2:** Clarification of the contribution of this paper is needed. In my opinion, this paper provided a good technical toolkit for retrieval, processing and analyzing of keywords related to transboundary water conflict and cooperation, which is a highly specific target. A "methodological framework" (between line 50-60) claimed by the authors, requires inclusions of multiple principles and theoretical components that serve a broader goal. That being said, I think it's also more suitable to move the implication of the toolkit (line 60-75) in the Introduction Section to Section 4.

**Authors Response#2:** We appreciate this suggestion from the reviewer that a broader goal should be addressed in the Introduction Section. In the revised manuscript, we will add a brief description of principles underpinning the logic of the proposed framework, which will be followed by a broader scope of its implications in the summary section.

The theory that inspired our framework, especially the logic of the Keyword Generator is from Lasswell's communication theory, who focused on communication as a process, to conduct problem-oriented inquiry of the news report through content analysis with the seven fundamental elements "who, with what intentions, in what situations, with what assets, using what strategies, reaches what audiences, with what result?". Our Search Keywords Generator flow chart shown in Figure 2 follows closely to the line of theoretical principles by Lasswell and intends to track conflict and cooperation dynamics on transboundary rivers by answering Lasswell' question involved with seven elements.

We thank the reviewer for raising this concern of confusion and will rewrite the Introduction Section to make it clearer in the revised manuscript. Sentences concerning the broader goal will be added in the Introduction Section, and technical details regarding to the toolkit will be simplified and partly moved to the Summary Section.

**Authors' Response to Referees' Comments**

**RC#3:** There are numerous data sources that collect news media data. What media sources are covered (print and web news? Other social media platforms such as Twitter?) Also, can the authors provide more details on how the LexisNexis data source is input in the toolkit and possibly other data sources? How does this toolkit perform in integrating data collected from multiple data sources?

**Authors Response#3:** Thank you for this comment. Choice of media sources should accord closely with the research goal. Our research goal is to track conflict and cooperation dynamics on transboundary rivers, which requires the data to cover water events and public opinion in a relatively long period of time. Also, to ensure data quality, newspapers (both print news and web news) written by professional journalists and editors are more suitable to take as data sources to reflect opinions of communities rather than social media (e.g. Twitter) as reflections of individual opinions.

LexisNexis contains the long-term coverage of mainstream newspapers and serves as one of the most commonly-used news media databases in the field of social sciences. Therefore, LexisNexis is chosen as an example of News Media Data Source and other suitable databases can, of course, be feasible options, as discussed in Section 2.1.1.

For data integration, any news data downloaded from suitable data sources (not only from LexisNexis) can be arranged and structured in the format of Table 4 through dada cleaning and processing procedure, stated in the Section of 2.3. After data processing, the toolkit provided by this research can be applied to the integrated data regardless of the original sources of it.

**RC#4:** In Table 1 the authors summarized the special treatments about basin names. Can you provide more explanations on how to determine the frequency setting when treating river basins with the same names but located in different continents? Also, regarding Block 5, are there any principles to determine what key words should be excluded?

**Authors Response#4:** Thank you for this comment. Usually when people talk about transboundary water issues, they focus on interactions on the scale of local communities and riparian states rather than intercontinental, and do not refer to the *Continent* name. Therefore, raising the frequency of continent name in search keywords will only compress data volume of relevant articles significantly, but not improve the data relevance pertaining to the research goal. However, river basins with the same names but located in different continents have different riparian countries. Adding frequency setting of riparian countries will filter out articles about the river on the other continent effectively. For example, St. John rivers appear both in Africa (flowing through Côte d'Ivoire, Guinea, and Liberia) and North America (flowing through the U.S and Canada). Rising frequency of riparian countries rather than continent names contributes more to the data relevance.

For the Excluded Terms shown in Block 5, given the research goal of our study, most of the terms are adopted from TFDD (Yoffe & Larson, 2001). These terms, seemingly relevant to our topics, occur in media articles massively and easily bring in lots of data noise. For example, 'sea' and 'ocean' bring mass of irrelevant articles talking about

marine rights and navigational utilization; 'nuclear' refers to 'nuclear power' and 'nuclear threaten', which is not the main concern of transboundary water conflict and cooperation; and as for 'flood of refugees', though it contains the keyword 'flood', but is regarded as irrelevant to our topics.

If researchers employ our framework in their own study fields in the future, Excluded Terms to avoid noise in Block 5 should be adopted to fit their own research field according to results of trial-and-error stated in Line 88 between Step 2 and Step3 and combined with their experience and knowledge background.

**RC#5:** Section 3.1.3 and Figure 5: this section is not clear to me. What is the data used to generate Figure 5, all 286 rivers? 60 Key rivers? Or just Lake Chad as it appears to have the greatest frequency? And more explanation is needed on why presenting the word clouds in both titles and text body.

**Authors Response#5:** Thanks for the comments. Data used in Section 3.1.3 is the dataset of 60 Key Research Basins. The concern to generate Word Clouds in both titles and text bodies is presenting differences of contents implied by both two. The result that Lake Chad appears to have the greatest frequency is just the direct and simple reflections of data. Analysis and explanation of this result is beyond the scope of this study. To avoid confusion for readers, Figure 5 will be deleted in the revised manuscript.

**RC#6:** Line 315-335: there are too many useless explanations of the figures such as "vertical axis represents…horizontal axis represents…" These are already clearly illustrated on the figures and does not need to be stated in words again.

**Authors Response#6:** We appreciate this suggestion from the reviewer and will simplify the explanations of the figures in the paper.

**RC#7:** The whole paper needs to be grammatically checked again.

**Authors Response#7:** We appreciate this suggestion from the reviewer and will check the whole paper again to improve the writing proficiency.

**Authors' replies to Comments of Reviewer #2:**

**RC#8:** This study proposed a framework to extract news articles related to transboundary rivers from a large newspaper database and demonstrated the application of the framework. It is a timely topic and relevant to the topics concerned by HESS. However, I have several concerns regarding the quality of this study.

**Authors Response#8:** We appreciate reviewer #2 for his/her constructive suggestions and comments. We agree to re-work on the main issues pointed out by the reviewer to progress the manuscript further. Our explanations and responses to all the reviewer's comments and questions are listed below.

**RC#9:** My major concern is on the significance of this study. The study constructed a framework to retrieve news texts related to transboundary rivers from a large database.

**Authors' Response to Referees' Comments**

The framework primarily relies on the term generator proposed by the authors to filter the news articles in the database, and the term generator is essentially a dictionary mapping of related terms. From my perspective, this study neither developed new cutting-edge techniques nor applied any advanced text mining techniques to resolve water resources issues, it looks more like a data preparation section of another paper rather than a standalone paper. In particular, the method is primarily based on term-based filtering, which is simple and straight-forward, and does not contribute much to current methodological studies; and the final product is a news database related to transboundary rivers, which only provides unstructured text data and does not directly provide any additional information, insights or solutions to any existing water resources issues. I would suggest the authors to work on two directions: (1) further develop more structured database through extracting more information from the original news texts with advanced text mining tools; (2) develop a few relatively simple cases to demonstrate the use of the data (i.e., implement some of the potential analysis.)

**Authors Response#9:** Thank you for this comment. As stated in Abstract and Introduction Section, tracking of conflict and cooperation dynamics on transboundary rivers are fundamental for better understanding of transboundary water resources management. News media has been considered as a valid proxy to track the evolving dynamics of societal value/public opinion over a longitudinal timeframe. However, existing studies mainly employed manual coding and sorting method to read and clean the data, which is time and labor consuming that limit its further applications in the era of big data. Departing from this perspective, this study tries to establish a methodological framework to generate news media datasets in a global scale for this research goal and provide a potent research toolkit to make it possible to save manual efforts sharply.

This paper is a standalone technical note, which provides simple and straight-forward but useful method to directly generate news media datasets tracking of conflict and cooperation dynamics on transboundary rivers. Also, the framework proposed in this paper possesses extensibility and compatibility to other research topics besides transboundary water resources management. Search Keywords Generator shown in Figure 2 follows closely Lasswell' question involved with seven elements- "who, with what intentions, in what situations, with what assets, using what strategies, reaches what audiences, with what result?" (stated in AR#2 in details). Figure 1 Method flow chart and Figure 2 Search Keywords Generator flow chart can be adopted for other research topics.

Shown in Table 4, the datasets are structured and provide good data preparation for potential analysis stated in Section 2.4. The scope of this paper is to demonstrate the effectiveness of the framework and toolkit, which is presented in Section 3 Results.

**RC#10:** In addition to the major concern above, I also have a few minor concerns as listed below. (1) The authors put too much potential impacts of this study as the significance of this study. I think the authors should clearly state what are the contributions of their work and what are the potential impacts.

**Authors Response#10:** We appreciate this suggestion from the reviewer that

clarification of contribution of this paper and potential analysis is further needed. As stated in AR#9, the main contribution of this paper is building a methodological framework and toolkit for news media dataset tracking of conflict and cooperation dynamics on transboundary rivers in a global scale, which saves manual efforts to a large extent. Shown in Figure 1 (also in Section 2.4), Potential Analysis is an important component of our framework, which demonstrates the utilization of our framework. According to this comment, we will make clarifications in Introduction Section to avoid confusion in the future.

**RC#11:** (2) There are many subject judgements in the workflow of the framework proposed by the authors, for example, the determination of the terms and 5 blocks, the determination of "satisfactory keyworks" in line 88, the manual relevance checking. what is "the balance between relevance and coverage" in line 137, etc. How can the authors ensure the subject judgement are not biased?

**Authors Response#11:** We thank the reviewer for this comment. As for manual relevance checking, we employed two manual coders to work independently, who had passed the training and inter-coder reliability test stated in Line 351-356. Krippendorff's Alpha-Reliability test is a common measure to avoid manual reading bias.

Term frequency setting of keywords is crucial to enhance data relevance. We use trial-and-error method to find the acceptable settings for our study. Compared to manual screening results of Lancang-Mekong River Basin without frequency setting (after filtering out irrelevant articles by manual reading), results after frequency setting provides nearly equivalent number of relevant articles before manual reading.

Balance between relevance and coverage refers that neither too much relevant information is missed, nor too much irrelevant information is included. Also, as stated previously, justification is based on trial-and-errors between Step 2 and Step 3.

To clarify the trail-and-error process we conducted to determine the frequency setting, we will update in the revised manuscript with the process details.

We will provide a series of tables to demonstrate the effects of various groups of frequency settings of keywords and how balance between relevance and coverage is approaching as appendix in the revised manuscript. Our term frequency settings of keywords and justification of balance between relevance and coverage may not be optimal, with a certain degree of coexisting subjectivity and objectivity. But they can serve as a reference for other researchers. As for other research topis, trial-and-errors between Step 2 and Step 3 are still needed, as also stated in AR#4.

**RC#12:** (3) In section 2, the potential analysis is listed as one part of the workflow, but it is only talked on the conceptual level. It is to some extent misleading to be listed as one step of the framework.

**Authors Response#12:** We thank the reviewer for this comment. As responded previously in AR#10, we will make clarifications in Introduction Section in the revised

manuscript to avoid confusion in the future.

**RC#13:** (4) In line 111, the author mentioned only English newspapers are collected. I wonder that whether it will cause some bias. For example, if there are two countries along a transboundary river, but English is the official language of only one of the two countries; then the collected news will be unbalanced, and the contents may only reflect the comments from one country.

**Authors Response#13:** We acknowledge that one of the limitations of this study is that only English newspapers were retrieved and included for analysis, which we might miss a variety of local languages newspaper sources that representing the local voices and perspectives. For future research, this could be improved by covering local languages through multiple newspaper databases, as stated in Line 401.

However, English, one of UN official languages, is usually used to disseminate opinions to international community even in non-English spoken countries. That is to say, English only news media can be used to reflect the viewpoints of riparian countries on the transboundary water management issues, which has been well demonstrated in the published paper on this HESS special issue (see Wei et al., 2021).

In addition, this study chose English as an example language to demonstrate the effectiveness of the proposed framework. We believe that researchers who focus on a specific riparian country in a river basin can choose their own local languages to generate their own news media datasets using this framework.

**RC#14:** (5) Part of the study is developed based on TFDD, e.g., line 128, line 191 and line 186. Please discuss the difference between your work and TFDD, and what is improved.

**Authors Response#14:** We thank the reviewer for this comment. Stated in Line 13-22, TFDD is built by means of manual reading for information extraction, thus difficult for fast updating, also does not cover the global changes in the past decade. Mentioned in AR#9, to fulfill the knowledge gap, this study builds a framework to generate news media datasets at a global scale for this research goal and provide a potent research toolkit to make it possible to save manual efforts sharply. This is the significance of this study.

We adopted the basic search keywords from TFDD and further revised to include five blocks of terms (Line 128) to make it extensible and adjustable according to a certain research topic. Through Block 1 and Block 2 with corresponding toolkit (Figure 2), a dataset covering transboundary rivers in a global scale can be generated, which is improved than TFDD. All the special treatments for basin names (shown in Table 1), country names (shown in Block2), and term frequency setting of keywords (stated in Section 2.2.3) are crucial measures to enhance data quality and save manual efforts, which are improvement beyond TFDD.

We will make further clarification of the improvements that our study has made compared with TFDD in the revised manuscript. Thanks.

**Authors' Response to Referees' Comments**

**RC#15:** (6) Line 206, how "5 time" is determined? Line 211, how to revise term frequency? Only based on subjective judgement?

**Authors Response#15:** We appreciate this comment from the reviewer. Referring to AR#11, we will provide a series of tables to demonstrate the effects of various groups of frequency settings of keywords and how balance between relevance and coverage is approaching as appendix in the revised manuscript. Our term frequency settings of keywords may not be optimal, with a certain degree of coexisting subjectivity and objectivity. But they can serve as a reference for other researchers. As for other research topics, trial-and-errors between Step 2 and Step 3 are still needed, as also stated in AR#4.

**RC#16:** (7) Line 221, please state clearly how the database sort the news articles by relevance, since you used the sorting function several times.

**Authors Response#16:** 'Sort by Relevance' is one of the sorting functions provided by LexisNexis, stated in Line 221. They also provide 'Sort by Date' and 'Sort by Document Title'. Among the three options, 'Sort by Relevance' works best for us to read roughly to change the frequency setting of keywords by trial-and-error. Therefore, we choose 'Sort by Relevance' before downloading the data from LexisNexis. Usually, every news database has the option for readers to sort by relevance.

**RC#17:** (8) Line 225-227. Only meta data are structured. It will be more helpful if the unstructured contents of the news can be somehow structured.

**Authors Response#17:** We appreciate this comment from the reviewer. Data structuring has its own applicable scene. This paper focuses on the process of building datasets. Our goal is the basic step of data structuring in which data is standardized into a tabular format with numerous rows and columns, making it easier to store and process for further analysis. Therefore, reserving the original data of whole articles can adapt to more potential analytic measures. On the basis of this dataset, further structuring, sentiment analysis and topic analysis etc. can be applied according to the researcher's goal.

**RC#18:** (9) Line 252, where LDA is used?

**Authors Response#18:** We thank the reviewer for this comment. Actually, LDA is a popular algorithm of topic modeling analysis. As stated in AR#17, further analysis of the data is in next paper. We will rewrite this paragraph and delete this sentence in the revised manuscript to avoid confusion.

**RC#19:** (10) As shown in your case study (figure 6), for some river basins the results are not acceptable (e.g., Columbia.) Have you evaluated how many occurrences of such regions in all your retrieved data? Do you have any measures to control the quality of the data?

**Authors Response#19:** We thank the reviewer for this comment. The purpose of Figure 6 is to demonstrate the necessity to apply special treatments for some river basins. Shown in Table 1, Columbia falls into the categories of basins which need special

treatments (since Columbia is both a district's name and a commercial brand). It makes sense that the data quality of Columbia River Basin is not as good as others. That is why special treatments are needed for basins mentioned in Table 1. Treatments needed corresponding to various categories of basins are also listed in Table 1.

**RC#20:** (11) In addition to transboundary rivers, is there any broader impacts of your study to the field of water resources and hydrology?

**Authors Response#20:** We appreciate this comment by the reviewer. As stated in AR#9, the framework proposed in this paper possesses extensibility and compatibility to other research topics besides transboundary water resources management.

**RC#21:** (12) Too many unnecessary details are provided in the major content of the paper. I would suggest the authors to write the main texts concisely.

**Authors Response#21:** We agree with the reviewer and will be more concise when providing details in the revised manuscript and will move part of details into appendix if necessary. With the help of this comment, we will re-arrange and rewrite part of the Data and Method Section and Results Section.

---

## Author Response (AR1)

**Final Authors' Response**

We would like to thank both Reviewers for their constructive comments on how to improve the manuscript. In the revised manuscript, updates are colored in red for easy reading. In the following, we provide answers to all the comments on a point-to-point basis. Each answer is structured as follows: (1) **RC#** comments from Referees, (2) **AR#** author's response.

**Authors' replies to Comments of Reviewer #1:**

**RC#1:** This paper develops a toolkit to track media data on conflict and cooperation events in transboundary river basins. It is a praising approach to provide systematic collection and mining of mass qualitative data in transboundary river basin management. The paper provided detailed procedures on how such toolkit should be implemented, and further illustrated its applicability with various case rivers. It was presented in a well-structured manner, however, there are some concerns that needed to be addressed.

**Authors Response#1:** We would like to thank reviewer #1 for the positive comments, which we believe will help to improve the manuscript substantially. We have reworked on the main issues pointed out by the reviewer to progress the manuscript further. Our explanations and responses to all the reviewer's comments and questions are listed below.

**RC#2:** Clarification of the contribution of this paper is needed. In my opinion, this paper provided a good technical toolkit for retrieval, processing and analyzing of keywords related to transboundary water conflict and cooperation, which is a highly specific target. A "methodological framework" (between line 50-60) claimed by the authors, requires inclusions of multiple principles and theoretical components that serve a broader goal. That being said, I think it's also more suitable to move the implication of the toolkit (line 60-75) in the Introduction Section to Section 4.

**Authors Response#2:** We appreciate this suggestion from the reviewer that a broader goal should be addressed in the Introduction Section. In the revised manuscript, we have added a brief description of principles underpinning the logic of the proposed framework, which will be followed by a broader scope of its implications in the summary section, updated in Abstract Section Line 17-21 and in Introduction Section Line 58-64.

More specifically, the theory that inspired our framework, especially the logic of the Keyword Generator is from Lasswell's communication theory, who focused on communication as a process, to conduct problem-oriented inquiry of the news report through content analysis with the seven fundamental elements "who, with what intentions, in what situations, with what assets, using what strategies, reaches what audiences, with what result?". Our Search Keywords Generator flow chart shown in Figure 2 follows closely to the line of theoretical principles by Lasswell and intends to track conflict and cooperation dynamics on transboundary rivers by answering Lasswell' question involved with seven elements. Updates of the above-mentioned principles could be found in Introduction Section Line 58-64.

Sentences concerning the broader goal have been updated in the Introduction Section, and the implications of the toolkit have been summarized after its description and thus have been move to the Summary Section by following the suggestion, updated in Line 416-424.

**RC#3:** There are numerous data sources that collect news media data. What media sources are covered (print and web news? Other social media platforms such as Twitter?) Also, can the authors provide more details on how the LexisNexis data source is input in the toolkit and possibly other data sources? How does this toolkit perform in integrating data collected from multiple data sources?

**Authors Response#3:** Thank you for this comment. Choice of media sources should accord closely with the research goal. Our research goal is to track conflict and cooperation dynamics on transboundary rivers, which requires the data to cover water events and public opinion in a relatively long period of time. Also, to ensure data quality, newspapers (both print news and web news) written by professional journalists and editors are more suitable to take as data sources to reflect opinions of communities rather than social media (e.g., Twitter) as reflections of individual opinions. To avoid confusion of future readers we have updated details in the revised manuscript in Section 2.1.1 Line 98-101.

Lexis Advance contains the long-term coverage of mainstream newspapers and serves as one of the most commonly-used news media databases in the field of social sciences. Therefore, Lexis Advance is chosen as an example of News Media Data Source and other suitable databases can, of course, be feasible options, as discussed in Section 2.1.1. Line 113-116.

For data integration, any news data downloaded from suitable data sources (not only from Lexis Advance) can be arranged and structured in the format of Table 4 through dada cleaning and processing procedure, stated in the Section of 2.3. After data processing, the toolkit provided by this research can be applied to the integrated data regardless of the original sources of it. Explanations are updated in Section 2.3 Line 253-256.

**RC#4:** In Table 1 the authors summarized the special treatments about basin names. Can you provide more explanations on how to determine the frequency setting when treating river basins with the same names but located in different continents? Also, regarding Block 5, are there any principles to determine what key words should be excluded?

**Authors Response#4:** Thank you for this comment. Usually when people talk about transboundary water issues, they focus on interactions on the scale of local communities and riparian states rather than intercontinental, and do not refer to the *Continent* name. Therefore, raising the frequency of continent name in search keywords will only compress data volume of relevant articles significantly, but not improve the data relevance pertaining to the research goal. However, river basins with the same names but located in different continents have different riparian countries. Adding frequency setting of riparian countries will filter out articles about the river on the other continent

effectively. For example, St. John rivers appear both in Africa (flowing through Côte d'Ivoire, Guinea, and Liberia) and North America (flowing through the U.S and Canada). Rising frequency of riparian countries rather than continent names contributes more to the data relevance. Clarifications are also updated in Section 2.2.2 Line 157-164.

For the Excluded Terms shown in Block 5, given the research goal of our study, most of the terms are adopted from TFDD (Yoffe & Larson, 2001). These terms, seemingly relevant to our topics, occur in media articles massively and easily bring in lots of data noise. For example, 'sea' and 'ocean' bring mass of irrelevant articles talking about marine rights and navigational utilization; 'nuclear' refers to 'nuclear power' and 'nuclear threaten', which is not the main concern of transboundary water conflict and cooperation; and as for 'flood of refugees', though it contains the keyword 'flood', but is regarded as irrelevant to our topics. Explanations are updated in Section 2.2.2 Line 205-210.

If researchers apply our framework in their own study fields in the future, Excluded Terms to avoid noise in Block 5 should be adopted to fit their own research field according to results of trial-and-error stated in Line 88 between Step 2 and Step3 and combined with their experience and knowledge background. Explanations are updated in Section 2.2.2 Line 211-213.

**RC#5:** Section 3.1.3 and Figure 5: this section is not clear to me. What is the data used to generate Figure 5, all 286 rivers? 60 Key rivers? Or just Lake Chad as it appears to have the greatest frequency? And more explanation is needed on why presenting the word clouds in both titles and text body.

**Authors Response#5:** Thanks for the comments. Data used in Section 3.1.3 is the dataset of 60 Key Research Basins. The initiative to generate Word Clouds in both titles and text bodies is presenting differences of contents implied by both. The result that Lake Chad appears to have the greatest frequency is just the direct and simple reflections of data. Given that analysis and explanation of this result is beyond the scope of this study, Figure 5 has been deleted in the revised manuscript to avoid confusion for readers.

**RC#6:** Line 315-335: there are too many useless explanations of the figures such as "vertical axis represents…horizontal axis represents…" These are already clearly illustrated on the figures and does not need to be stated in words again.

**Authors Response#6:** We appreciate this suggestion from the reviewer and have simplified the explanations of the figures in the paper.

**RC#7:** The whole paper needs to be grammatically checked again.

**Authors Response#7:** We appreciate this suggestion from the reviewer and have checked the whole paper again to improve the writing proficiency.

**Final Authors' Response**

**RC#8:** This study proposed a framework to extract news articles related to transboundary rivers from a large newspaper database and demonstrated the application of the framework. It is a timely topic and relevant to the topics concerned by HESS. However, I have several concerns regarding the quality of this study.

**Authors Response#8:** We appreciate reviewer #2 for his/her constructive suggestions and comments. We have reworked on the main issues pointed out by the reviewer to progress the manuscript further. Our explanations and responses to all the reviewer's comments and questions are listed below.

**RC#9:** My major concern is on the significance of this study. The study constructed a framework to retrieve news texts related to transboundary rivers from a large database. The framework primarily relies on the term generator proposed by the authors to filter the news articles in the database, and the term generator is essentially a dictionary mapping of related terms. From my perspective, this study neither developed new cutting-edge techniques nor applied any advanced text mining techniques to resolve water resources issues, it looks more like a data preparation section of another paper rather than a standalone paper. In particular, the method is primarily based on term-based filtering, which is simple and straight-forward, and does not contribute much to current methodological studies; and the final product is a news database related to transboundary rivers, which only provides unstructured text data and does not directly provide any additional information, insights or solutions to any existing water resources issues. I would suggest the authors to work on two directions: (1) further develop more structured database through extracting more information from the original news texts with advanced text mining tools; (2) develop a few relatively simple cases to demonstrate the use of the data (i.e., implement some of the potential analysis.)

**Authors Response#9:** Thank you for this comment. As stated in Abstract Line 8-12 and Introduction Section Line 33-35, tracking of conflict and cooperation dynamics on transboundary rivers are fundamental for a better understanding of transboundary water resources management. News media has been considered as a valid proxy to track the evolving dynamics of societal value/public opinion over a longitudinal timeframe. However, existing studies mainly employed manual coding and sorting method to read and clean the data, which is time and labor consuming that limit its further applications in the era of big data. Departing from this perspective, this study tries to establish a methodological framework to generate news media datasets in a global scale for this research goal and provide a potent research toolkit to make it possible to save manual efforts sharply.

This paper is a standalone technical note, which provides simple, straight-forward but useful method to directly generate news media tracking of conflict and cooperation dynamics on transboundary rivers. Also, the framework proposed in this paper possesses extensibility and compatibility to other research topics besides transboundary water resources management. Principles underpinning Search Keywords Generator shown in Figure 2 follows closely Lasswell's question involved with seven elements-"who, with what intentions, in what situations, with what assets, using what strategies,

reaches what audiences, with what result?" (stated in AR#2 in details). Figure 1 Method flow chart and Figure 2 Search Keywords Generator flow chart can be adopted for other research topics. Explanations are also updated in revised manuscript in Summary Section Line 417-424 and Line 426-427.

Shown in Table 4, the datasets are structured and provide good data preparation for potential analysis stated in Section 2.4. The scope of this paper is to demonstrate the effectiveness of the framework and toolkit, which is presented in Section 3 Results.

**RC#10:** In addition to the major concern above, I also have a few minor concerns as listed below. (1) The authors put too much potential impacts of this study as the significance of this study. I think the authors should clearly state what are the contributions of their work and what are the potential impacts.

**Authors Response#10:** We appreciate this suggestion from the reviewer that clarification of contribution of this paper and potential analysis is further needed, and have stated clearer in Summary Section Line 417-424. As stated in AR#9, the main contribution of this paper is building a methodological framework and toolkit for news media dataset tracking of conflict and cooperation dynamics on transboundary rivers in a global scale, which saves manual efforts to a large extent. Shown in Figure 1 (also in Section 2.4), Potential Analysis is an important component of our framework, which demonstrates the utilization of our framework. According to this comment, we have made clarifications in Abstract Section Line 19-21 and deleted further statement of potential impacts in Line 94 in the revised manuscript to avoid confusion in the future.

**RC#11:** (2) There are many subject judgements in the workflow of the framework proposed by the authors, for example, the determination of the terms and 5 blocks, the determination of "satisfactory keyworks" in line 88, the manual relevance checking. what is "the balance between relevance and coverage" in line 137, etc. How can the authors ensure the subject judgement are not biased?

**Authors Response#11:** We thank the reviewer for this comment. The study does involve certain amount of manual efforts for relevance and coverage checking, however, the subjective words used in the manuscript were all supported by pre-defined rules and test to ensure reliability and accuracy. For example, during the manual relevance checking process, two coders, who were trained beforehand, were involved to work independently, results coded by these two coders were then undergone inter-coder reliability test (Krippendorff's Alpha-reliability test) to avoid biases among coders. More details regarding the test could be found in Line 374-379.

The trial-and-error process conducted to determine keywords term frequency setting followed the same principle and process. In the revised manuscript, we have further clarified such justifications behind what we meant by "balance between relevance and coverage", updates can be found in revised manuscript in Appendix. The results include recordings of trial-and-error processes to demonstrate the effects of various groups of frequency settings of keywords and how balance between relevance and coverage is approached.

**RC#12:** (3) In section 2, the potential analysis is listed as one part of the workflow, but it is only talked on the conceptual level. It is to some extent misleading to be listed as one step of the framework.

**Authors Response#12:** We thank the reviewer for this comment. We have updated the workflow figure (Figure 1) accordingly in the revised manuscript to avoid future confusion and updated corresponding statement in Section 2 Line 86, 93 and 258.

**RC#13:** (4) In line 111, the author mentioned only English newspapers are collected. I wonder that whether it will cause some bias. For example, if there are two countries along a transboundary river, but English is the official language of only one of the two countries; then the collected news will be unbalanced, and the contents may only reflect the comments from one country.

**Authors Response#13:** We acknowledge that one of the limitations of this study is that only English newspapers were retrieved and included for analysis, which we might miss a variety of local languages newspaper sources that representing the local voices and perspectives. For future research, this could be improved by covering local languages through multiple newspaper databases, as stated in Line119-121 and Line 432-433.

However, English, one of UN official languages, is usually used to disseminate opinions to international community even in non-English spoken countries. That is to say, English only news media can be used to reflect the viewpoints of riparian countries on the transboundary water management issues, which has been well demonstrated in the published paper on this HESS special issue (see Wei et al., 2021).

In addition, this study chose English as an example language to demonstrate the effectiveness of the proposed framework. We believe that researchers who focus on a specific riparian country in a river basin can choose their own local languages to generate their own news media datasets using this framework.

**RC#14:** (5) Part of the study is developed based on TFDD, e.g., line 128, line 191 and line 186. Please discuss the difference between your work and TFDD, and what is improved.

**Authors Response#14:** We thank the reviewer for this comment. The major difference between this study and TFDD lies in the design of keywords, which allows efficient retrieval of targeted search. We built on the basic search keywords from TFDD but further revised to include five blocks of terms (Line 137) to make it extensible and adjustable for other research topics. In particular, the special treatments for basin names (shown in Table 1), riparian country names (shown in Block2), and term frequency setting of keywords (stated in Section 2.2.3) are crucial measures to enhance data quality and save manual efforts.

**RC#15:** (6) Line 206, how "5 time" is determined? Line 211, how to revise term frequency? Only based on subjective judgement?

**Authors Response#15:** We appreciate this comment from the reviewer. Referring to

AR#11, we have provided appendix to demonstrate the effects of various groups of frequency settings of keywords and how balance between relevance and coverage is approaching in the revised manuscript.

**RC#16:** (7) Line 221, please state clearly how the database sort the news articles by relevance, since you used the sorting function several times.

**Authors Response#16:** 'Sort by Relevance' is one of the sorting functions provided by Lexis Advance. They also provide 'Sort by Date' and 'Sort by Document Title'. Among the three options, 'Sort by Relevance' works best for us to read roughly to change the frequency setting of keywords by trial-and-error. Therefore, we choose 'Sort by Relevance' before downloading the data from Lexis Advance. Usually, every news database has the option for readers to sort by relevance. To avoid future confusion, we have added more details in Line 244-248.

**RC#17:** (8) Line 225-227. Only meta data are structured. It will be more helpful if the unstructured contents of the news can be somehow structured.

**Authors Response#17:** We appreciate this comment from the reviewer. Data structuring has its own applicable scene. This paper focuses on the process of building datasets. Our goal is the basic step of data structuring in which data is standardized into a tabular format with numerous rows and columns, making it easier to store and process for further analysis. Therefore, reserving the original data of whole articles can adapt to more potential analytic measures. On the basis of this dataset, further structuring, sentiment analysis and topic analysis can be applied according to the researcher's goal.

**RC#18:** (9) Line 252, where LDA is used?

**Authors Response#18:** We thank the reviewer for this comment. Actually, LDA is a popular algorithm of topic modeling analysis. As stated in AR#17, further analysis of the data is in next paper. We have rewritten this paragraph and deleted this sentence in the revised manuscript to avoid confusion.

**RC#19:** (10) As shown in your case study (figure 6), for some river basins the results are not acceptable (e.g., Columbia.) Have you evaluated how many occurrences of such regions in all your retrieved data? Do you have any measures to control the quality of the data?

**Authors Response#19:** We thank the reviewer for this comment. The purpose of Figure 6 is to demonstrate the necessity to apply special treatments for some river basins. Shown in Table 1, Columbia falls into the categories of basins which need special treatments (since Columbia is both a district's name and a commercial brand). It makes sense that the data quality of Columbia River Basin is not as good as others. Explanations are updated in Section 3.2 Line 397-404. That is why special treatments are needed for basins mentioned in Table 1. Treatments needed corresponding to various categories of basins are also listed in Table 1.

**RC#20:** (11) In addition to transboundary rivers, is there any broader impacts of your

study to the field of water resources and hydrology?

**Authors Response#20:** We appreciate this comment by the reviewer. As stated in AR#9, the framework proposed in this paper possesses extensibility and compatibility to other research topics besides transboundary water resources management, also updated in Line 426-427.

**RC#21:** (12) Too many unnecessary details are provided in the major content of the paper. I would suggest the authors to write the main texts concisely.

**Authors Response#21:** We have simplified details in the revised manuscript and moved part of details into appendix. We have rearranged and rewritten part of the Data and Method Section and Results Section.

---

## Referee Report (RR1)

I am mostly satisfied with the authors' responses to my comments, with the following comments to be addressed:

Line 19: Please specify clearly the scope/coverage of data analysed and the dataset created.

Line 57-82: This section is still not clear as it mixes the aim of the research with its implications. The detailed aims and a short, more concise implication should be clearly stated in this introduction section, whereas more detailed implications should be moved to the summary section at the end of the article.

Line 397: I suppose you are referring to Figure 5 here?

---

## Author Response (AR2)

**Authors' Response to Referees' Comments**

We would like to thank both Reviewers for their efforts and patience in reviewing process and for constructive comments on how to improve the manuscript. In the revised manuscript, updates are colored in red for easy reading. In the following, we provide answers to all the comments on a point-to-point basis. Each answer is structured as follows: (1) **RC#** comments from Referees, (2) **AR#** author's response.

**Authors' replies to Comments of Reviewer #1:**

**RC#1:** I am mostly satisfied with the authors' responses to my comments, with the following comments to be addressed: Line 19: Please specify clearly the scope/coverage of data analysed and the dataset created.

**Authors Response#1:** We would like to thank reviewer #1 for the positive comments, which we believe will help us to further improve the manuscript. We have reworked on the main issues pointed out by the reviewer to progress the manuscript.

In the abstract section, we have specified the coverage of data more clearly in Line 19-20. The continental coverage of 60 Key Research Basins is Asia, North America, Africa, Europe and South America, except Antarctica and Oceania, updated in Line 316. Note that Figure 3 shows spatial coverage in basin scale and covers 286 transboundary river basins in the world; however, Figure 4 shows spatial coverage in country scale in the world. Therefore, in Figure 3, Oceania has no data since there is no transboundary river basins located in Oceania; and in Figure 4 Australia has a certain amount of data since it has released news media papers on other transboundary rivers internationally. Updated in Line 19, the temporal coverage of the datasets of 60 Key Research Basins is from 1953 to 2019; in Line 20, the relevance screening is conducted on the four representative river basins, also stated in Sect 3.2.

**RC#2:** Line 57-82: This section is still not clear as it mixes the aim of the research with its implications. The detailed aims and a short, more concise implication should be clearly stated in this introduction section, whereas more detailed implications should be moved to the summary section at the end of the article.

**Authors Response#2:** We appreciate this suggestion from the reviewer that detailed implications should be moved to the summary section on the basis of the previous version of revised manuscript. In this revised manuscript, we have added a brief description of implications of this research in Line 68-69, and moved detailed implications from Line 70-85 to the summary section in Line 433 to 450.

**RC#3:** Line 397: I suppose you are referring to Figure 5 here?

**Authors Response#3:** Thank you for this comment. The in-text reference of Figure 5 has been updated in Line 400.

**There are no further comments from Reviewer #2 in this round of review.**